

# Connecting the Greenland ice-core and U/Th timescales via cosmogenic radionuclides: Testing the synchronicity of Dansgaard-Oeschger events

Florian Adolphi[1,2], Christopher Bronk Ramsey[3], Tobias Erhardt[1], R. Lawrence Edwards[4], Hai Cheng[4,5], Chris S. M. Turney[6], Alan Cooper[7], Anders Svensson[8], Sune O. Rasmussen[8], Hubertus Fischer[1] and Raimund Muscheler[2]

[1]Climate and Environmental Physics, Physics Institute & Oeschger Centre for Climate Change Research, University of Bern, Sidlerstrasse 5, 3012 Bern, Switzerland
[2]Quaternary Sciences, Department of Geology, Lund University, Sölvegatan 12, 22362 Lund, Sweden
[3]Research Laboratory for Archaeology and the History of Art, University of Oxford, Dyson Perrins Building, South Parks Road, Oxford OX1 3QY, UK
[4]Insitute of Global Environmental Change, Xi'an Jiatong University, Xi'an 710049, China
[5]Department of Erath Sciences, University of Minnesota, Minneapolis, Minnesota, 55455, USA
[6]Palaeontology, Geobiology and Earth Archives Research Centre and ARC Centre of Excellence in Australian Biodiversity and Heritage, School of Biological, Earth and Environmental Sciences, University of New South Wales, Sydney, NSW 2052, Australia
[7]Australian Centre for Ancient DNA and ARC Centre of Excellence in Australian Biodiversity and Heritage, School of Biological Sciences, The University of Adelaide, Adelaide, SA 5005, Australia
[8]Centre for Ice and Climate, Niels Bohr Institute, University of Copenhagen, Juliane Maries Vej 30, 2100 Copenhagen, Denmark

*Correspondence to:* Florian Adolphi (adolphi@climate.unibe.ch)

**Abstract.** During the last glacial period Northern Hemisphere climate was characterized by extreme and abrupt climate changes, so-called Dansgaard-Oeschger (DO) events. Most clearly observed as temperature changes in Greenland ice-core records, their climatic imprint was geographically widespread. However, the temporal relation between DO-events in Greenland and other regions is uncertain due to the chronological uncertainties of each archive, limiting our ability to test hypotheses of synchronous change. On the contrary, the assumption of direct synchrony of climate changes forms the basis of many timescales. Here, we use cosmogenic radionuclides ($^{10}$Be, $^{36}$Cl, $^{14}$C) to link Greenland ice-core records to U/Th-dated speleothems, quantify offsets between both timescales, and improve their absolute dating back to 45,000 years ago. This approach allows us to test the assumption that DO-events occurred synchronously between Greenland ice-core and tropical speleothem records at unprecedented precision. We find that the onset of DO-events occurs within synchronization uncertainties in all investigated records. Importantly, we demonstrate that there remain local discrepancies in the temporal development of rapid climate change for specific events and speleothems. These may be either related to the location of proxy records relative to the shifting atmospheric fronts or to underestimated U/Th-dating uncertainties. Our study thus highlights the potential for misleading interpretations of the Earth system when applying the common practice of climate wiggle-matching.

## 1 Introduction

Precise and accurate chronologies are critical for understanding past environmental and climatic changes. Global natural and anthropogenic archives can only be directly compared through the development of robust



chronological frameworks, enabling studies of the spatiotemporal dynamics of past change. These findings are
crucial for understanding the nature and cause of rapid climate changes in the past, and hence, characterizing the
dynamics and feedbacks of past and projected future climate change (Thomas, 2016). However, the
applicability, precision, and accuracy of the available dating methods pose strong constraints on our ability to
infer leads and lags between climate records, and ultimately, mechanisms of change in the Earth system.
Instead, the situation is often reversed: climate changes such as Dansgaard-Oeschger, or DO, events (Dansgaard
et al., 1993; Dansgaard et al., 1969) are typically *assumed* to occur synchronously across the Northern
Hemisphere in different climate proxies from various regions and then used as chronological tie-points. This so-
called "climate wiggle-matching" forms the chronological basis of a large part of paleoclimate records (e.g.,
Bard et al., 2013; Hughen et al., 2006; Henry et al., 2016; Turney et al., 2015), especially in the marine realm
where other dating methods suffer from low precision and poorly constrained biases such as the marine
radiocarbon reservoir age (Lougheed et al., 2013). Furthermore, it also plays a central role for one of the most
widely used dating methods in paleosciences – the radiocarbon dating method. About one third of the data
underlying the current radiocarbon calibration curve, IntCal13 (Reimer et al., 2013), obtain their absolute age
from climate wiggle-matching.
Climate wiggle-matching has the obvious drawback that the leads and lags between different climate records
cannot be studied once the records have been forced to align. The approach critically rests on the assumptions,
that i) the climate change indeed occurred synchronously everywhere, and that ii) the (sometimes fundamentally
different) proxies in question record the changes in a similar way and without intrinsic delays. These
assumptions, however, can very rarely be rigorously tested but when they are, ample evidence is revealed that
questions their universal validity. Lane et al. (2013) showed that rapid climate change in the North Atlantic
region may be time transgressive with regional leads and lags on the order of a century. Nakagawa et al. (2003)
argued that the onset of Greenland Interstadial 1e (GI-1e, Rasmussen et al., 2014a) occurred multiple centuries
after the associated climate shift in Japan (and subsequent revisions of the underlying timescales (Staff et al.,
2013; Bronk Ramsey et al., 2012; Seierstad et al., 2014) did not resolve this conundrum). Buizert et al. (2015)
inferred that the Southern Ocean response to DO-events is delayed by about 200 years on average while the
atmosphere around Antarctica reacted instantaneously (Markle et al., 2016). Baumgartner et al. (2014) found
asynchronicities between ice-core proxies for local Greenland temperature ($\delta^{15}$N) and the tropical/mid-latitude
hydrological cycle ($CH_4$) during some DO-events. They discussed that the climate changes in polar and low-
latitude regions may indeed be synchronous, but that atmospheric $CH_4$ concentrations rise with a delay during
some DO-events because of compensating changes in the source strengths of the northern and southern
hemisphere wetlands. Alternatively, their findings can be explained via a real delay between Greenland climate
change and the activation of $CH_4$ source areas during certain DO-events. Fleitmann et al. (2009) reported on
timing differences of DO-events in Greenland ice cores and speleothems, albeit largely within dating
uncertainties. However, they also found significant differences between speleothem records outside their
chronological uncertainties. This is complemented by a recent study showing that the duration of a stadial-
interstadial transition can differ by up to 300 years between different East Asian speleothems (Li et al., 2017)
emphasizing the questions of whether we should expect the onset, mid-point, or end-point of DO-events to
occur simultaneously, as this choice will lead to different results when aligning the records.



In this paper, we attempt to provide improved constraints on the paradigm of climate synchronicity. We employ
cosmogenic radionuclides as a climate-independent synchronization-tool between the Greenland ice-core
timescale (Andersen et al., 2006; Rasmussen et al., 2006; Seierstad et al., 2014; Svensson et al., 2008; Svensson
et al., 2006; Vinther et al., 2006) and the U/Th timescale (Broecker, 1963; Edwards et al., 1987; Cheng et al.,
2013a) and reduce the absolute dating error of the Greenland ice cores by 50 – 70% back to 45,000 years BP
(Before Present, 1950). This allows us to compare the timing of DO-type variability seen in key paleoclimate
records at unprecedented precision: The Greenland ice cores and U/Th-dated (sub-)tropical speleothems.
**2 Cosmogenic radionuclides as synchronization tools**
Cosmogenic radionuclides (such as $^{14}$C, $^{10}$Be and $^{36}$Cl) are produced in a nuclear cascade that is triggered when
galactic cosmic rays (GCR) collide with the Earth's atmosphere's constituents (Lal and Peters, 1967). While the
GCR flux outside the heliosphere can be assumed to be constant over the past million years (Vogt et al., 1990),
the flux arriving at Earth is modulated by the strength of the helio- and geomagnetic fields (Masarik and Beer,
1999). Hence, the production rates of cosmogenic radionuclides are inversely related to changes in solar activity
and/or the strength of the geomagnetic field. This modulation effect leaves a globally synchronous, externally
forced signal in cosmogenic radionuclide records around the world. Hence, they can serve as a powerful
synchronization tool for climate archives from different regions. The challenge lies in estimating potential non-
production-related impacts on radionuclide concentrations in a given archive that may result from geochemical
and meteorological processes.
After production, $^{14}$C is oxidized to $^{14}$CO$_2$ and enters the carbon cycle. Changing $^{14}$C production rates thus alter
the atmospheric $^{14}$C/$^{12}$C ratio (expressed as per mille $\Delta^{14}$C, that is, $^{14}$C/$^{12}$C corrected for fractionation and decay
relative to a standard, denoted $\Delta$ in Stuiver & Pollach, 1977). Due to carbon cycle effects, these variations in
$\Delta^{14}$C are dampened and delayed with respect to the causal production rate changes (Siegenthaler et al., 1980;
Roth and Joos, 2013). In addition to variable production rates, changes in the exchange rates between the
different carbon pools can alter $\Delta^{14}$C. The world's oceans in particular have a significantly lower $\Delta^{14}$C than the
contemporary atmosphere due to their long carbon residence time (Craig, 1957). Thus, variations in the $^{14}$C
exchange rates between the ocean and the atmosphere will alter atmospheric $\Delta^{14}$C independent of production
rate changes.
$^{10}$Be attaches to aerosols and is transported from the stratosphere to the troposphere within 1-2 years (Raisbeck
et al., 1981) mainly via mid-latitude tropopause breaks (Heikkilä et al., 2011). It has no active geochemical
cycle and so its atmospheric concentration is a more direct recorder of production rate changes compared with
$\Delta^{14}$C. However, $^{10}$Be transport and deposition in the troposphere is guided by local meteorology and thus
susceptible to changes thereof (Heikkilä and Smith, 2013; Pedro et al., 2011). This can cause variations in $^{10}$Be
records that are not related to production rate changes. Furthermore, a so-called "polar bias" (i.e., an
overrepresentation of polar as opposed to global production rate changes) has been proposed for ice-core records
(Bard et al., 1997). This would lead to subdued geomagnetic and enhanced solar modulation of ice-core
radionuclide records due to the geometry of the geomagnetic field. However, there is no consensus in different
empirical studies and modelling experiments to whether this effect is present and the results may also vary





regionally (Bard et al., 1997; Heikkilä et al., 2009a; Pedro et al., 2012; Adolphi and Muscheler, 2016;
Muscheler and Heikkilä, 2011; Field et al., 2006).
The transport and deposition of $^{36}$Cl in its aerosol phase is comparable to $^{10}$Be. However, in addition to an
aerosol phase, $^{36}$Cl also has a gaseous phase (H$^{36}$Cl) which is likely dominant in the stratosphere (Zerle et al.,
1997). In the troposphere, the partitioning between aerosol and gas phase is not well understood.  It may vary in
space and time (Lukasczyk, 1994), and can change rapidly depending on pH (Watson et al., 1990). The gaseous
H$^{36}$Cl phase can also be lost from acidic ice in low accumulation sites after deposition which is, however, less
relevant for the high accumulation sites studied here (Delmas et al., 2004). In Greenland, similar to $^{10}$Be, the
dominant deposition process of $^{36}$Cl in is wet deposition (Heikkilä et al., 2009b) which is supported by the
overall similarity of $^{36}$Cl and $^{10}$Be variations recorded in ice cores (Wagner et al., 2001b; Muscheler et al.,

126    2005).

As a result, all three radionuclides depend on the same production mechanism which causes their production
rates to co-vary globally. This signal can be exploited for global synchronization of paleorecords from natural
archives. However, to identify these common changes, their different geochemistry needs to be accounted for. In
the case of radiocarbon this is achieved through carbon cycle modelling, to deconvolve the effects of the carbon
cycle on the relation between $^{14}$C production rates and $\Delta^{14}$C (Muscheler et al., 2004). For $^{10}$Be and $^{36}$Cl, fluxes
can be calculated from ice accumulation rates. This provides a first-order correction for changing
paleoprecipitation rates on the ice sheet and their influence on the radionuclide concentrations. In reality, aerosol
transport to the ice sheet is more complex and depends on changes in transport velocity, pathways and
scavenging effects en route (Schüpbach et al., 2018), which are, however, difficult to constrain for $^{10}$Be due to
its stratospheric origin. Instead, comparisons of fluxes and concentrations to other climate proxies can inform
about potential climate influences on $^{10}$Be and$^{36}$Cl transport and deposition (Adolphi and Muscheler, 2016). It is
currently not possible to quantitatively correct either of the radionuclides for these non-production influences
since neither past carbon cycle changes nor atmospheric circulation changes are sufficiently well known.
However, the potential amplitude of non-production rate changes can be assessed through sensitivity
experiments and added as an uncertainty for the production rate signal (Adolphi and Muscheler, 2016; Köhler et
al., 2006).
The potential of this synchronization tool has been demonstrated multiple times to infer differences between the
tree-ring and ice-core timescales (Adolphi and Muscheler, 2016; Muscheler et al., 2014a; Southon, 2002), test
the accuracy of the radiocarbon calibration curve (Adolphi et al., 2017; Muscheler et al., 2014b; Muscheler et
al., 2008), and synchronize ice cores from both hemispheres (Raisbeck et al., 2017; Raisbeck et al., 2007).
**3 Methods & Data**
**3.1 Ice-Core Data**
The ice-core $^{10}$Be and $^{36}$Cl data used in this study are shown in figure 1. We focus on records that have been
robustly linked to the GICC05 timescale (Andersen et al., 2006; Rasmussen et al., 2006; Seierstad et al., 2014;
Svensson et al., 2008; Rasmussen et al., 2008). Hence, the majority of the data stems from the deep Greenland
ice cores GRIP, GISP2, and NGRIP. In addition, we use Antarctic $^{10}$Be fluxes from EDC, EDML and Vostok



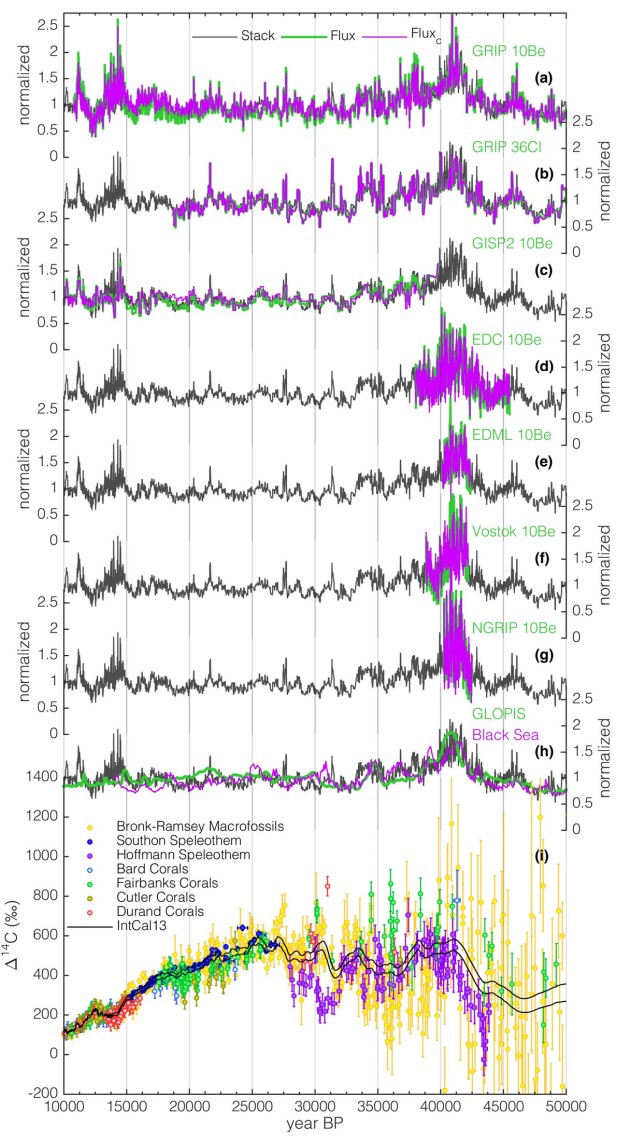

**Figure 1: Data used in this study. Panel a-g show individual ice-core records of GRIP [10]Be (Baumgartner et al., 1997b; Muscheler et al., 2004; Wagner et al., 2001a; Yiou et al., 1997; Adolphi et al., 2014), GRIP [36]Cl (Baumgartner et al., 1998; Baumgartner et al., 1997a; Wagner et al., 2001b; Wagner et al., 2000), GISP2 [10]Be (Finkel and Nishiizumi, 1997), and [10]Be from EDC, EDML, Vostok, and NGRIP (all Raisbeck et al., 2017). Each record represents deposition fluxes (green) and 'climate corrected' fluxes (purple, see text). In addition, each panel contains the stack of all ice-core records (black, see text). Panel h: [10]Be production rates modelled from two geomagnetic field intensity reconstructions: GLOPIS (green, Laj et al., 2004) and based on Black Sea sediments (purple, Nowaczyk et al., 2013) using the production rate model by Herbst et al. (2016). The ice-core radionuclide stack is shown in black. All records in panel a-h are shown on the GICC05 timescale (Seierstad et al., 2014) and normalized to (i.e., divided by) their mean. Panel i: Absolutely dated [14]C data from Lake Suigetsu (yellow, Bronk Ramsey et al., 2012), Hulu Cave (blue, Southon et al., 2012), Bahamas speleothems (purple, Hoffmann et al., 2010), and various tropical coral datasets (Bard et al., 1998; Cutler et al., 2004; Durand et al., 2013; Fairbanks et al., 2005, shown in light blue, olive, red, and green, respectively). The black lines encompass the ±1σ uncertainties of IntCal13 (Reimer et al., 2013).**



that have been anchored to GICC05 by matching solar variability present in all $^{10}$Be records, and volcanic tie-
points (Raisbeck et al., 2017).
By calculating fluxes we make a first order correction for the changing snow accumulation rates between
stadials and interstadials and their influence on radionuclide concentrations (Wagner et al., 2001b; Johnsen et
al., 1995; Rasmussen et al., 2013; Finkel and Nishiizumi, 1997). The accumulation rates for each ice core are
based on their annual layer thickness – derived from their individual timescales – corrected for ice thinning. For
the Greenland ice cores this thinning function is based on a 1-D ice flow model (Dansgaard and Johnsen, 1969;
Johnsen et al., 1995; Johnsen et al., 2001; Seierstad et al., 2014). For the Antarctic ice cores we use the strain
rate derived from the Bayesian ice-core dating effort AICC12 (Veres et al., 2013). These strain rates are
inherently uncertain and independently derived accumulation rate estimates differ by up to 10-20% in the glacial
(Gkinis et al., 2014; Rasmussen et al., 2013; Guillevic et al., 2013). However, these differences are largely
systematic and change only on multi-millennial timescales. The shorter term changes in accumulation rates are a
more direct function of the timescale, which is very precise for increments of the core (Rasmussen et al., 2006).
This is important to note, as we mainly exploit production rate changes on centennial to millennial timescales
for synchronization.
To test for additional climate influences on $^{10}$Be or $^{36}$Cl deposition in the ice cores, we followed the approach by
Adolphi and Muscheler (2016): For each ice core we calculated multiple linear regression models using $\delta^{18}$O
and snow accumulation rates as predictors for $^{10}$Be ($^{36}$Cl) fluxes and subtracted the obtained model from the
$^{10}$Be ($^{36}$Cl) data. We denote the resulting record as the "climate corrected flux" (Flux$_c$). This approach may
correct climate effects on $^{10}$Be ($^{36}$Cl) deposition insufficiently, or it may over-correct them, so it cannot be
assumed per se that the resulting record is more reliable than the original fluxes. Nevertheless, it provides a first
order sensitivity test for the ice-core records with respect to climate-related transport and depositional effects on
$^{10}$Be ($^{36}$Cl) fluxes.
To combine all ice-core records, we calculated their mean (denoted as "Stack", Fig. 1) using Monte-Carlo
bootstrapping (Efron, 1979). Using 7 ice-core records in two versions (flux and flux$_c$) yields a total number of
14 samples. In each iteration, 14 samples are randomly drawn (with replacement, i.e., each record can be drawn
multiple times), perturbed within measurement errors, and stacked. Repeating this procedure 1,000 times we
obtain an average relative standard deviation of 8% between the derived stacks, which is comparable to the
measurement uncertainty of individual measurements but larger than the expected error of the mean which
points to systematic differences between the records. For the period where we have data from both hemispheres
this standard deviation is only slightly higher (10%). Even though this is only a relatively short period (see Fig.
1), it contains multiple DO-events which are expressed differently in Northern and Southern Hemisphere
climate. Thus, this agreement can serve as indication that climate effects do not dominate the signal.
**3.2 Radiocarbon data**
For the purpose of this study we have to focus on radiocarbon records that are absolutely dated. Furthermore,
the length and sampling resolution of the records need to be sufficient to resolve centennial-to-millennial
production rate changes. The records that fulfil these criteria are shown in figure 1 and comprise $^{14}$C data from
various U/Th dated coral records (Bard et al., 1998; Durand et al., 2013; Cutler et al., 2004; Fairbanks et al.,
2005), as well as $^{14}$C measured in two speleothems (Southon et al., 2012; Hoffmann et al., 2010). In addition,



we use the [14]C record from Lake Suigetsu (Bronk Ramsey et al., 2012) since the U/Th dated records do not
directly reflect atmospheric [14]C but the ocean mixed layer (corals) and, in the case of speleothems, a mixture of
atmospheric and soil $CO_2$, and carbonate bedrock from above the cave. The timescale of the Lake Suigetsu
record has been inferred from matching its [14]C record to the [14]C variations in speleothems, additionally
constrained by varve counting (Bronk Ramsey et al., 2012). Hence, it is not truly independently dated. However,
similar to ice-core layer counting, this varve count adds constraints especially on centennial timescales, so that
$\Delta$[14]C variations on these timescales should be relatively unaffected by this tuning to the speleothem [14]C data.
Thus, even though the timescale may not be independent, this record can still be used to verify the existence of
$\Delta$[14]C variations in the atmosphere seen in the mixed layer records.
In addition, we use the available tree-ring records back to 14,000 cal BP in the revised version by Hogg et al.
(2016)(not shown in figure 1 for clarity).

### 3.3 Carbon cycle modelling

To be able to compare ice-core and radiocarbon records directly we have to account for the effects of the carbon
cycle. Following earlier studies (Muscheler et al., 2004; Muscheler et al., 2008), we use a box-diffusion carbon
cycle model (Siegenthaler et al., 1980) to model $\Delta$[14]C from the ice-core radionuclide records. We assume that
ice-core [10]Be ([36]Cl) variations are proportional to [14]C production rate changes and model $\Delta$[14]C anomalies from
each realization of the ice-core stack, as well as the single ice-core records (Fig. 2). It can be seen that the
modelled $\Delta$[14]C records from the individual ice-core records differ in their long-term trends since the carbon
cycle integrates over time so that relatively small but systematic differences in the radionuclide fluxes (possibly
arising from uncertainties in the strain rates) have a significant effect on longer time scales. However, all records
show the same overall evolution of $\Delta$[14]C. Furthermore, especially when subtracting the long-term trend and
isolating variations on timescales shorter than 5000 years, the agreement is very high (on average within 15‰ at
$1\sigma$, Fig. 2b), which is the part of the signal that we will be exploiting in our synchronization effort.

### 3.3.1 Production rate ratio

Modeling $\Delta$[14]C values from [10]Be measurements is based on the assumption that [10]Be and [14]C production rate
changes are proportional to each other. However, different production rate models differ in their sensitivity of
[14]C and [10]Be production rate changes to variations in the geomagnetic field (Cauquoin et al., 2014). For a given
geomagnetic field change, the production rate model by Masarik and Beer (2009, 1999) yields 30-50% lower
[10]Be production rate changes than the calculations by Poluianov et al. (2016) and Herbst et al. (2016). For [14]C on
the other hand, all models yield roughly similar amplitudes. This leads to differences in the [14]C/[10]Be production
rate ratio for a given change in the geomagnetic field. If Masarik and Beer (1999) are correct, the variations in
ice-core [10]Be records have to be upscaled by 30-50% to be proportional to [14]C production rate changes while no
such scaling is necessary when the other production rate models are used. In addition, the amplitudes in [14]C and
[10]Be may differ due to the presence of polar bias (see section 2). If this effect was present, then geomagnetic
field changes should cause bigger variations in [14]C than [10]Be.
Since the presence of a polar bias is debated and the physical reason for the differences between the production
rate models is unresolved, we chose an empirical approach to scale the ice-core record appropriately:





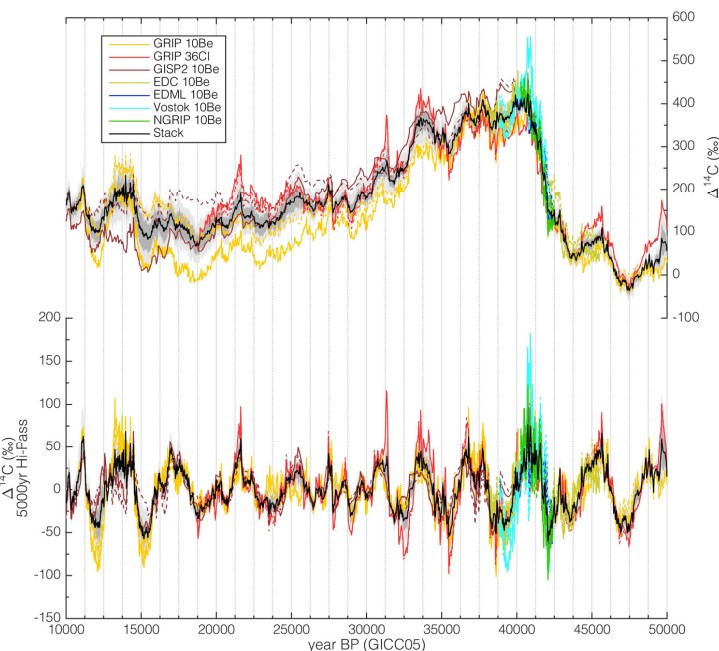

**Figure 2: Modelled $\Delta^{14}C$ anomalies from individual ice-core records (see legend, solid lines are based on radionuclide fluxes while dashed lines are inferred from flux$_c$) and the realizations of the ice-core stack (black line shows the mean of all realizations, dark and light grey shading encompass 68.2 and 95.4% probability ranges). The top panel shows the unfiltered model output. The bottom panel displays the records after variations with frequencies <1/5000a$^{-1}$ have been subtracted (FFT-based filter).**

We use three geomagnetic field intensity reconstructions around the Laschamp geomagnetic field minimum (Laj et al., 2004; Laj et al., 2000; Nowaczyk et al., 2013) and calculate the resulting $^{10}$Be production rate changes using the production rate models by Masarik and Beer (1999) and Herbst et al. (2016) (Fig. 3 a-c). Subsequently, we scale the ice-core $^{10}$Be record to minimize the root mean square error (RMSE) between ice-core and geomagnetic field-based records (Fig. 3d). It can be seen that the RMSE reaches a minimum for a $^{10}$Be scaling factor of ~1 (for Masarik and Beer, 1999) and ~1.3 (for Herbst et al., 2016). This represents a fortunate coincidence; irrespective of which production rate model is used, the amplitude of the ice-core $^{10}$Be variations has to be increased by approximately 30% to match $^{14}$C. If the production rate model by Masarik and Beer is used, then the amplitude of the ice-core $^{10}$Be record is in agreement with geomagnetic field data, but due to the higher production sensitivity of $^{14}$C (see above), $^{10}$Be variations have to be increased by ~30%. Similarly, if the production rate model by Herbst et al. is used, then the amplitude of the ice-core $^{10}$Be record is 30% smaller than implied by geomagnetic field data (possibly due to a polar bias), while the sensitivity of $^{14}$C and $^{10}$Be is the same. Again, the net effect is the $^{10}$Be variations have to be scaled up by 30% for the comparison to $^{14}$C.





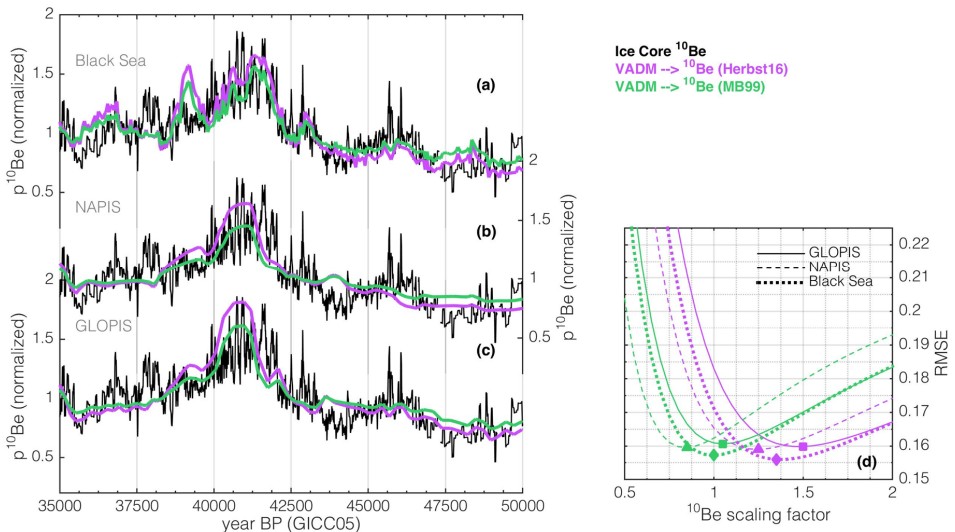

**Figure 3: Comparison of ice-core-based and geomagnetic-field-based reconstructions of $^{10}$Be production rates. Panel a-c show the ice-core stack (black) in comparison to $^{10}$Be production rates based on geomagnetic field reconstructions and 2 different production rate models (Herbst et al. (2016) in pink and Masarik and Beer (1999) in green). Panel a, the Black Sea geomagnetic field record (Nowaczyk et al., 2013), Panel b, the NAPIS geomagnetic field stack (Laj et al., 2000), and Panel c, the GLOPIS geomagnetic field stack (Laj et al., 2004). Panel d shows the RMSE between the ice-core data and the geomagnetic-field-based records when variations in the ice-core record are scaled by different factors (x-axis). The colours correspond to the production rate models. The line styles indicate the geomagnetic field records (see legend) and the symbols denote the RMSE minima.**

### 3.3.2 The state of the carbon cycle

As mentioned in section 2, a quantification of transient carbon cycle changes and their influence on $\Delta^{14}$C is challenged by insufficient knowledge of inventories and processes. The contribution of single processes to $\Delta^{14}$C changes over the last glacial cycle is likely within 30‰ and, due to compensating effects, also their combination is likely not bigger than 40‰ (Köhler et al., 2006). Here we use the Laschamp event to estimate the state of the ocean ventilation around 40 ka BP.

The datasets underlying IntCal13 all show an increase of about 320‰ in $\Delta^{14}$C into the Laschamp event (Fig. 4), albeit at different absolute levels (see Fig. 1). This is ~100‰ more than the compiled IntCal13 curve itself implies. This disagreement can be explained by differences in timing and absolute $\Delta^{14}$C between the different datasets leading to smoothing and dampening of $\Delta^{14}$C variations during the construction of IntCal13. Also, geomagnetic field changes yield a $\Delta^{14}$C change more in line with the individual $^{14}$C datasets than with IntCal13, even when assuming a preindustrial carbon cycle.

To estimate the mean state of the carbon cycle during this period, we run our carbon cycle model with different (constant) values of ocean diffusivity. We find that modelled and measured $\Delta^{14}$C around the Laschamp event match best in amplitude when we run the model under conditions where ocean ventilation is reduced to ~75% of its preindustrial value (Fig. 4). This is in broad agreement with previous modelling experiments (Köhler et al., 2006; Roth and Joos, 2013) and proxy data (Henry et al., 2016).

In the following, we will use this estimate for the parameterization of our model. As mentioned above, a transient adjustment of carbon cycle parameters is uncertain and will hence not be attempted. Instead, we ascribe an associated uncertainty to the modelled $\Delta^{14}$C based on the carbon cycle sensitivity experiments by



Köhler et al. (2006). Furthermore, it should be noted, that by only using (filtered) $\Delta^{14}$C anomalies as
synchronization targets, we i) avoid systematic carbon cycle influences on $\Delta^{14}$C levels, and ii) minimize
transient carbon cycle related changes in $\Delta^{14}$C (Adolphi and Muscheler, 2016).

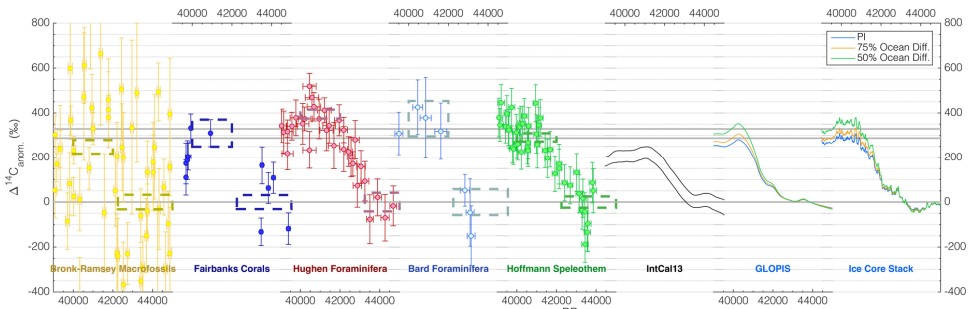


**Figure 4: The Laschamp event in measured and modelled $\Delta^{14}$C. The 6 panels to the left show $\Delta^{14}$C anomalies from**
**macrofossils from Lake Suigetsu (yellow, Bronk Ramsey et al., 2012), tropical corals (blue, Fairbanks et al., 2005),**
**foraminifera from Cariaco Basin sediments (red, Hughen et al., 2006), foraminifera from Iberian Margin sediments**
**(light blue, Bard et al., 2013), Bahamas speleothems (green, Hoffmann et al., 2010), and IntCal13 (black, Reimer et**
**al., 2013). The two panels on the right show modelled $\Delta^{14}$C using the GLOPIS (Laj et al., 2004) geomagnetic field**
**record as well as the ice-core stack as production rate inputs. The different coloured lines reflect different carbon**
**cycle scenarios (see legend, PI denotes pre-industrial). The conversion of geomagnetic field intensity to $^{14}$C production**
**rate is based on the production rate model by Herbst et al. (2016). Note, that the amplitude of the $^{10}$Be variations have**
**been increased by 30% as discussed in section 3.3.1.**

**3.4 Synchronization – effects of the carbon cycle and the archive**
The synchronization method follows Adolphi and Muscheler (2016) and is outlined and tested in detail therein.
In brief, sections of modelled (ice-core based) $\Delta^{14}$C anomalies are compared to the measured $\Delta^{14}$C. For our
analysis we employ high-frequency changes in $\Delta^{14}$C since carbon cycle changes have only limited effects on
atmospheric $\Delta^{14}$C on shorter time scales (Adolphi and Muscheler (2016). Similarly, as shown in figure 2, the
agreement of the different ice-core records is better on shorter timescales. In this study, we employ two types of
high pass filtering: a FFT-based high-pass filter and simple linear detrending. The choice of filter is based on the
data sampling resolution. For the highly resolved tree-ring data we use the FFT filter, while the lower resolved
and more unevenly sampled coral/speleothem/macrofossil data is filtered by linear detrending to avoid the
interpolation to equidistant resolution required for FFT analysis. The exact frequencies and window lengths are
given in the results section. Using the same statistics as for radiocarbon wiggle-match dating (Bronk Ramsey et
al., 2001), we then infer a probability density function (PDF) for the timescale difference between the modelled
and measured $\Delta^{14}$C records. For details of the statistics of this methodology we refer the reader to Adolphi and
Muscheler (2016). Here we focus instead on additional uncertainties that arise when comparing modelled
atmospheric $\Delta^{14}$C to $^{14}$C records from the ocean mixed layer (corals) or speleothems.
$\Delta^{14}$C variations in the atmosphere are dampened and delayed compared to the causal production rate changes.
Both factors, attenuation and delay, depend on the frequency of the production rate change (Roth and Joos,
2013; Siegenthaler et al., 1980). The dampening is largest at high frequencies and decreases with longer periods.
On the other hand, the apparent peak-to-peak delay between sinusoidal production rate changes and the resulting





$\Delta^{14}$C change is increasing with increasing wavelengths. Similar effects occur when comparing atmospheric and
oceanic $\Delta^{14}$C changes to each other: the ocean reacts to atmospheric $\Delta^{14}$C changes with a delayed and dampened
response that is wavelength dependent. Hence, we need to take these factors into account when comparing a
modelled atmospheric $\Delta^{14}$C record to mixed layer marine records. However, the frequency dependence of the
attenuation and delay makes it difficult to explicitly correct for this since atmospheric $\Delta^{14}$C changes vary on
different time scales simultaneously. Furthermore, the coral records vary in their sampling frequency and often
it is not precisely known over how much time an individual $^{14}$C sample integrates.
Figure 5 shows a sensitivity test regarding these effects. We modelled $\Delta^{14}$C from the ice-core stack around the
Laschamp event and compared the atmospheric $\Delta^{14}$C to the mixed layer $\Delta^{14}$C in the model. To simulate the
effect of varying averaging effects of the coral samples, we low-pass filtered the mixed layer signal with
increasing cut-off wavelengths. For each filter, we then inferred the apparent delay between the mixed layer
(i.e., the "coral") and the atmosphere. In doing so we infer that even though the signal is dominated by a long
lasting $\Delta^{14}$C increase, the inferred delay is small (~30 years) as long as the coral samples do not integrate over
long times. Only when assuming that each coral sample averages over more than 1,000 years we infer delays of
about 120 years. Nevertheless, this experiment also shows that within reasonable bounds of averaging, the delay
of mixed layer to atmospheric signal is limited

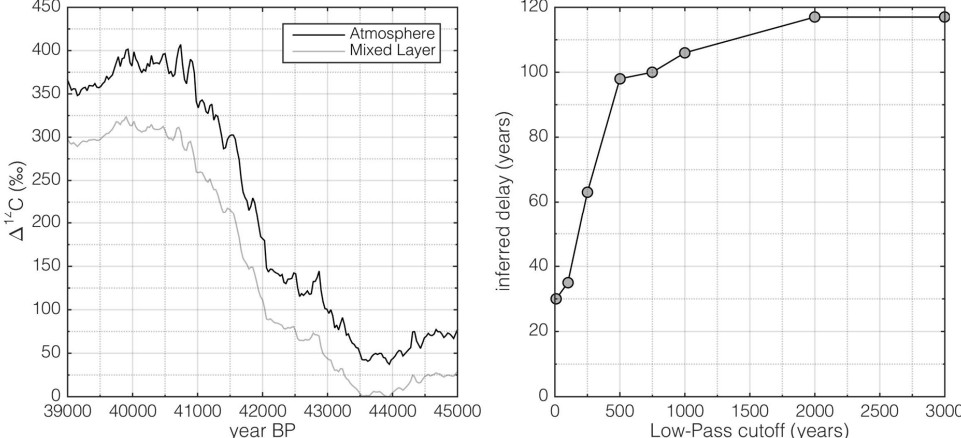

**Figure 5: The delay between $\Delta^{14}$C in the atmosphere and ocean mixed layer. The left panel shows modelled $\Delta^{14}$C**
**from the ice-core stack around the Laschamp event. The modelled atmospheric $\Delta^{14}$C is shown in black while ocean**
**mixed layer is shown in grey. They right hand panel shows the inferred delay from our synchronization method when**
**comparing the atmospheric to the mixed layer signal for different low-pass filters of the mixed layer signal (x-axis).**

The speleothem $\Delta^{14}$C reacts differently than the ocean mixed layer. The so-called dead carbon fraction (DCF) of
a speleothem consists of two main contributors: i) respired soil organic matter that is older (in $^{14}$C years) than
the atmospheric $^{14}$C signal, and ii) carbonate bedrock that contains no $^{14}$C. Applying the model of Genty and
Massault (1999), we model speleothem $\Delta^{14}$C using different assumptions on the age of the respired soil organic
matter and fraction of carbonate bedrock in drip water $CO_2$. We do this for 2 examples: i) a speleothem with an
apparent DCF (i.e., offset from the atmosphere) of 5.8% (resembling the Hulu Cave speleothem record by
Southon et al., 2012) and ii) a speleothem with an apparent DCF of 25.7% (resembling the Bahamas speleothem





by Hoffmann et al., 2010). By assuming different ages of the soil respired carbon (τ = 10 – 400 years, see Fig.
6), we adjust the fraction of $^{14}$C-free $CO_2$ so that the apparent DCF for each speleothem is matched. The age of
the soil respired carbon is defined following Genty and Massault (1999): if, for example, τ = 100 years, then the
activity of the soil respired $CO_2$ is the mean of the atmospheric activity over the past 100 years prior to sampling
(also accounting for decay within these 100 years). For simplicity we assume a uniform age distribution for the
soil respired carbon. Subsequently, we compare the modelled speleothem $\Delta^{14}$C to the original atmospheric input
using our synchronization method and plot the inferred delay (Fig. 6, right panel). From this experiment it can
be seen that the controlling factor on the inferred delay is the age of the soil respired matter that acts as an
integrator (low-pass filter) of the atmospheric $^{14}$C signal. The fraction of $^{14}$C-free carbonate has no influence on
the lag between $\Delta^{14}$C changes in the atmosphere and the speleothem, but only dampens the amplitude of the
corresponding change. Realistic ages of soil respired carbon differ from region to region but even though some
slow cycling fractions of soil organic matter may be up to several thousand years old (Trumbore, 2000), the
major contributors to soil $CO_2$ are considerably younger and in the order of decades (Genty et al., 2001;
Fohlmeister et al., 2011).
From these experiments we conclude that our systematic matching uncertainties to coral and speleothem records
are probably below 100 years. We note that this uncertainty is asymmetric since the ocean/speleothem signal
cannot lead the atmosphere and so the offset is unidirectional.

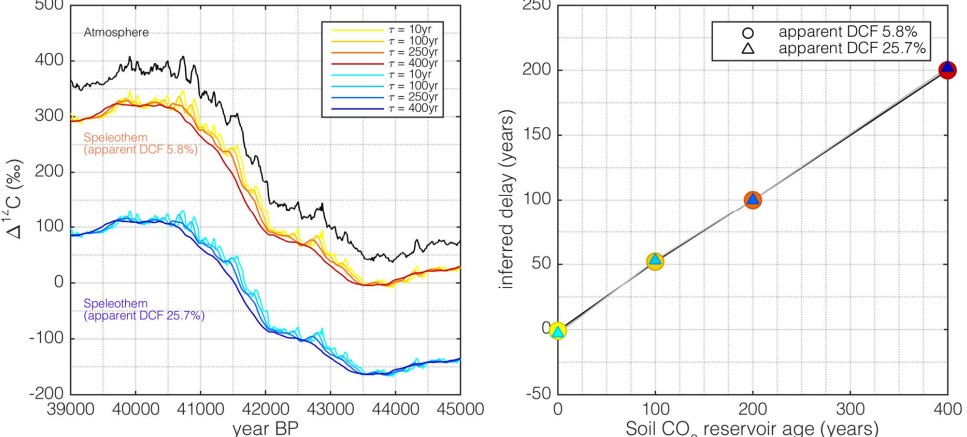


**Figure 6: Effect of varying ages of soil respired $CO_2$ and fractions of $CO_2$ from $^{14}$C-dead carbonate on the $\Delta^{14}$C in**
**speleothems. The left panel shows atmospheric modelled $\Delta^{14}$C from the $^{10}$Be stack (black) and two modelled**
**speleothem scenarios with a net DCF of 5.8% (warm colours) and 25.7% (cold colours). For each speleothem, a**
**number of different ages for the respired soil organic matter have been assumed (see legend) and the input of $^{14}$C-free**
**$CO_2$ from carbonate has been adjusted to obtain the correct apparent DCF value between 39-40.5 ka BP. The right**
**hand panel shows the inferred delay when we apply our synchronization method to match the atmospheric $\Delta^{14}$C to**
**the speleothem record.**
**3.5 Change-point detection in climate records**
To assess the synchronicity of abrupt climate changes between the climate records after the synchronisation we
use a probabilistic approach to detect their onsets in the individual records. The employed model describes the





abrupt changes as a linear transition between two constant states. To account for long-term fluctuations in the
climate records, deviations from the transition are described as autocorrelated noise. The approach and inference
procedure are described in more detail in Erhardt et al. (in prep). The model is fitted independently to windows
of the individual datasets (table 1 & Fig. 13) around the rapid transitions on their individual timescales. The
apparent delays between the inferred onsets of the transitions in the different records are then compared,
propagating the respective uncertainties. For each record, only events that are well expressed and measured in
high resolution have been fitted.
**Table 1. Change-point detection window for each record. For each investigated climate event and record, the change-**
**point detection algorithm has been applied between t1 and t2. The windows have been defined visually, ensuring a**
**sufficient amount of data prior to and after the transition. For each record, only events that are well expressed in the**
**climate proxy records at high resolution have been investigated. For the ice-core record t1 and t2 typically encompass**
**500 years prior to and after the nominal transition ages by Rasmussen et al. (2014a). The exact values have been**
**adjusted to exclude overlap with other transitions where necessary (Erhardt et al. in prep).**

| Event | GICC05 (yr BP) | MCE | Hulu d18O | | Sofular d18O | | Sofular d13C | | ElCondor d18O | | Diamante d18O | |
|---|---|---|---|---|---|---|---|---|---|---|---|---|
| | | | t1 | t2 | t1 | t2 | t1 | t2 | t1 | t2 | t1 | t2 |
| Holocene | 11653 | 99 | 12453 | 10503 | 12703 | 10703 | 12703 | 11003 | 12453 | 11203 | 13403 | 11203 |
| GI-1e | 14642 | 186 | 15442 | 13942 | 15442 | 13942 | 15442 | 13942 | 15442 | 14192 | 16392 | 14192 |
| GS-3 Dust Peak | 24130 | 645 | 25380 | 24080 | - | - | - | - | - | - | 25780 | 24630 |
| GI-3 | 27730 | 832 | 28580 | 27680 | 28780 | 27880 | 28780 | 27780 | - | - | 29030 | 28080 |
| GI-4 | 28850 | 898 | 30100 | 28900 | 30150 | 29400 | 30150 | 29200 | 29900 | 29000 | 30100 | 29100 |
| GI-5.1 | 30790 | 1008 | 31540 | 30790 | - | - | - | - | 31590 | 30740 | 32040 | 30840 |
| GI-5.2 | 32450 | 1132 | 33300 | 32200 | 33100 | 32400 | 33300 | 32200 | 33250 | 32000 | 33050 | 32450 |
| GI-6 | 33690 | 1195 | 34590 | 33640 | 34740 | 33690 | 34990 | 33540 | 34240 | 33490 | - | - |
| GI-7c | 35430 | 1321 | 36680 | 34980 | 36380 | 35480 | 36380 | 35230 | 36230 | 34880 | 36480 | 34980 |
| GI-8c | 38170 | 1449 | 39420 | 37420 | 39420 | 37220 | 39120 | 37220 | 39220 | 37220 | - | - |
| GI-9 | 40110 | 1580 | 40860 | 40060 | 40960 | 39960 | 41160 | 39960 | - | - | - | - |
| GI-10 | 41410 | 1633 | 42110 | 41060 | 42460 | 41590 | 42460 | 41460 | 42210 | 40960 | - | - |
| GI-11 | 43290 | 1736 | 44240 | 42940 | 44840 | 43540 | - | - | 44040 | 42440 | - | - |


## 4 Time scale differences between GICC05 and the U/Th timescale

In the following sections we will show the synchronization results for different time windows. We focus our
analysis on three distinct windows: 10-14 ka BP, 18-25 ka BP and 39-45 ka BP. The youngest window is
defined by the presence of high-resolution tree-ring data for [14]C back to 14 ka BP. Going further back in time it
becomes increasingly challenging to unequivocally identify common structures in the various $\Delta^{14}$C records that
are suitable for synchronization because the resolution of the individual records decreases back in time while
their differences to each other are growing steadily (see Fig. 1i). Hence, we focus on the well-known Laschamp
event around 41 ka BP, and the period between 18-25kaBP, i.e., preceding the major carbon cycle changes
associated with the deglaciation. We omit the period between 25-39 ka BP. As discussed in Reimer et al. (2013)
and seen in figure 1i there is substantial disagreement between the datasets underlying IntCal13 at that time that
are impossible to reconcile within their respective age and/or [14]C uncertainties. Hence, also any structure in the
$\Delta^{14}$C records may be unreliable and thus, lead to erroneous synchronization results.

### 4.1 10,000 – 14,000 years BP

In the 10-14 ka BP interval, we synchronize the ice-core stack to high-resolution tree-ring and speleothem $\Delta^{14}$C
data (Fig. 7). The high sampling resolution of the [14]C records allows us to focus on centennial-to-millennial





$\Delta^{14}C$ changes (<1000 years) where carbon cycle influences on $\Delta^{14}C$ can be expected to be small (Adolphi and
Muscheler, 2016). In concordance with earlier studies (Muscheler et al., 2014a) we find that GICC05 is ~65
years older than the tree-ring timescale at the onset of the Holocene, but that this offset vanishes over the course
of the Younger Dryas interval.
While Muscheler et al. (2014a) argued that this changing offset may be in part due to errors in the timescale of
the floating Late Glacial Pines, we can now support this change in the timescale-difference through the U/Th
dated speleothems: The synchronization of the ice-core stack to the H82 speleothem from Hulu Cave (Southon
et al., 2012) leads to fully consistent results as inferred from the tree-rings. This indicates that the most likely
explanation is an ice-core layer counting bias, i.e. that the GICC05 time scale suggests too old ages at the onset
of the Holocene, but is accurate within a few decades during GI-1.

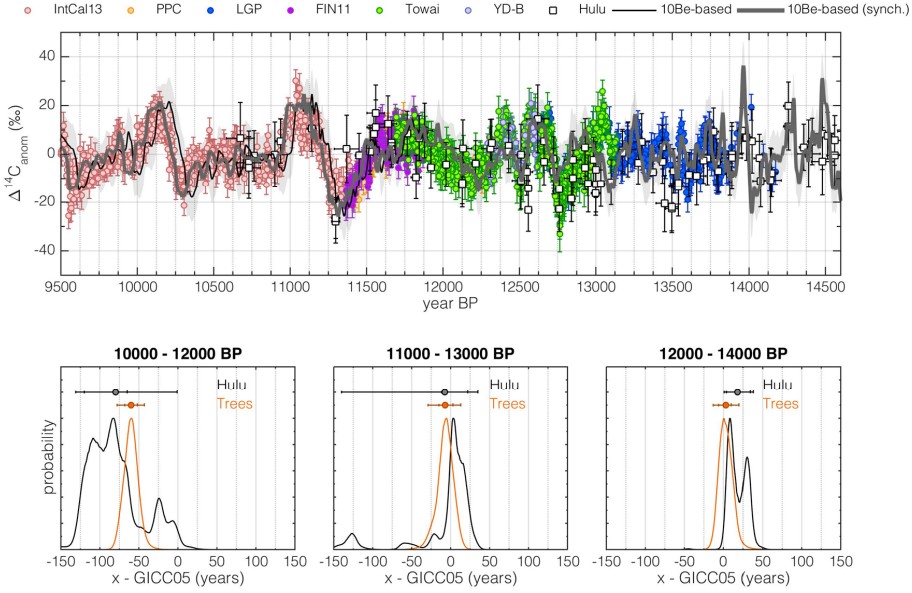


**Figure 7: Synchronization of GICC05 to tree-ring and Hulu Cave records during the last deglaciation. Top panel:**
**Ice-core based modelled $\Delta^{14}C$ anomalies on the original GICC05 timescale (thin black line, light grey shading**
**encompasses the ±10‰ uncertainty (±1σ) of the modelled $\Delta^{14}C$, based on the carbon-cycle sensitivity experiments by**
**Adolphi & Muscheler (2016)) and synchronized timescale (bold grey line). Tree-ring data underlying IntCal13 are**
**shown in pink. Revised Northern Hemisphere tree-ring data according to Hogg et al. (2016) are shown in orange**
**(Preboreal Pines), dark blue (Late Glacial Pine) and light blue (Younger Dryas-B chronology). New kauri $\Delta^{14}C$ data**
**by Hogg et al. (2016) is shown in purple (FIN11) and green (Towai). Hulu Cave H82 $\Delta^{14}C$ data are shown as white**
**squares. All symbols are shown with ±1σ error bars. All data are FFT-filtered to isolate $\Delta^{14}C$ variations on timescales**
**<1000 years. The lower three panels show inferred probability distributions of timescale differences between GICC05**
**and tree-rings (orange) and Hulu Cave (black). The symbols and error bars denote means, and 68.2% and 95.4%**
**confidence intervals of the inferred timescale difference. Each of the lower panels refers to a 2000-year subsection of**
**the data indicated at the top of each panel.**
Interestingly, we do not observe any significant differences between the results stemming from tree-rings and
the speleothem records. As shown in section 3.4, we could expect a delay in the speleothem $\Delta^{14}C$ compared to
the atmosphere if the respired soil organic carbon contribution to the soil $CO_2$ was very old. This would result in
GICC05 appearing older in comparison to the speleothem than relative to the tree rings. The lack of this delay
implies that the majority of the respired soil organic carbon at Hulu Cave must be younger than ~100 years (see
Fig. 6). This is supported by the fact that the centennial $\Delta^{14}C$ variations in the tree-ring and speleothem data



have the same amplitude (Fig. 7). If old organic carbon significantly contributed to the soil $CO_2$, we would
instead expect to see a stronger smoothing of short-term $\Delta^{14}C$ variations.

### 4.2 18,000 – 25,000 years BP

Due to the irregular and lower sampling resolution of the $^{14}C$ records beyond 15,000 cal BP, we chose to
linearly detrend each data set (instead of band-pass filtering) to remove offsets between the different $^{14}C$
datasets and highlight common variability. Furthermore, we have to increase the length of the comparison data
windows to 4,000 years to ensure sufficient structure in the $^{14}C$ sequences entering the comparison. Each
window is detrended separately in the analysis to isolate short-term $\Delta^{14}C$ variability. We note however, that
detrending each $^{14}C$ dataset over the entire timeframe (18-25 ka BP) instead does not alter the results
significantly. Compared to the high-frequency $\Delta^{14}C$ changes studied between 10-14 ka BP, the longer-term
variations used for synchronization here may have been increasingly affected by carbon-cycle changes. To
account for this, we increase the uncertainty estimate of the modelled $\Delta^{14}C$ changes to ±30‰ (±1σ), which is
sufficiently large to account for estimated carbon-cycle-driven $\Delta^{14}C$ changes from modelling experiments
during the entire glacial (Köhler et al., 2006). We note that this is a conservative estimate, given that during this
period neither modelling (Köhler et al., 2006; Muscheler et al., 2004), nor data (Eggleston et al., 2016) suggest
large carbon-cycle changes.

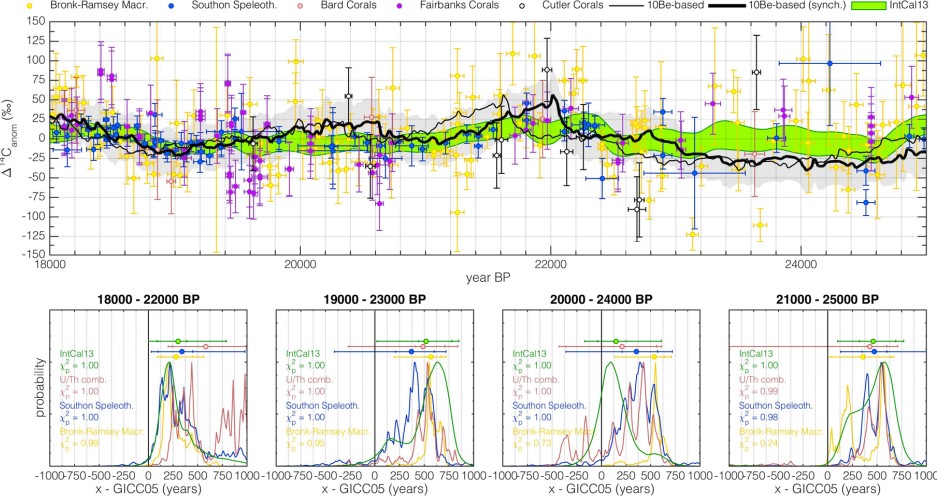


**Figure 8: Synchronization results between 18,000 and 25,000 years BP. Top panel: The thin black line shows the modelled $\Delta^{14}C$ curve based on the ice-core stack on its original timescale. The bold black line and grey shading show the synchronized ice-core record including assumed ±1σ uncertainties of ±30‰. The different coloured symbols indicate various $^{14}C$ datasets underlying IntCal13, which is shown as the green envelope. Lower panels: Each panel shows PDFs of the inferred timescale difference between the ice-core stack and IntCal13 (green), a combination of all U/Th-dated records (speleothems/corals, pink), the H82 speleothem (blue), and Lake Suigetsu (yellow). Symbols of similar colour show the inferred mean and 68.2% and 95.4% confidence intervals. Colour-coded text indicates $\chi^2$ probabilities for the goodness of fit between modelled and measured $\Delta^{14}C$ curves after synchronization. Small (e.g., <0.1) values would indicate significant disagreement. Note that all $\chi^2$ probabilities are relatively high, indicating that our uncertainty estimate for the modelled $\Delta^{14}C$ is very conservative. Each of the lower panels refers to a specific subsection of the data indicated at the top of each panel.**

It can be seen in figure 8 that it is challenging to infer robust co-variability in multiple $^{14}C$ records. However, the
millennial evolution of $\Delta^{14}C$ does show common changes in the 18-25 ka BP interval. Synchronizing the ice-





core stack to data from i) Hulu Cave H82 speleothem, ii) Lake Suigetsu macrofossils, iii) the IntCal13 stack or
iv) a combination of all U/Th dated records (speleothems/corals) leads to consistent results within uncertainties
for each choice of time windows: all records imply that GICC05 shows younger ages compared to the $^{14}$C
records around this time.
The most significant structure that is present in all measured and modelled $^{14}$C records during this time is the
centennial $\Delta^{14}$C increase around 22.1kaBP (see Fig. 9). Comparing the ice-core stack to $\Delta^{14}$C between 21-
23kaBP indicates an offset of ~550 years between GICC05 and the U/Th timescale around this time (GICC05
being younger). To account for the potential delay of coral and speleothem $\Delta^{14}$C compared to the atmosphere,
we also modelled the mixed layer $\Delta^{14}$C signal from the ice-core stack and synchronized this signal to the
measured $^{14}$C data (Fig. 9). As discussed in section 3.4, we find very little difference in the inferred timing since
the $\Delta^{14}$C variation is relatively rapid (centuries). Comparing the $\Delta^{14}$C anomalies to geomagnetic field data
shows that a small part of the longer-term development of this structure is probably driven by geomagnetic field
changes. The amplitude (~50‰) and short duration (centuries) of the $\Delta^{14}$C increase, however, suggest that this
change is mainly driven by a series of strong solar minima, comparable to the Grand Solar Minimum period
around the onset of the Younger Dryas (Muscheler et al., 2008).

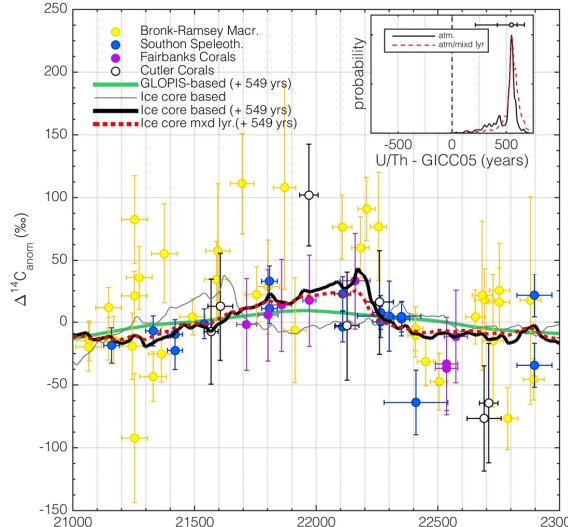


**Figure 9: Close-up of measured and modelled $\Delta^{14}$C anomalies between 21 and 23 ka BP. The thin grey line shows**
**modelled atmospheric $\Delta^{14}$C from the ice-core stack on the GICC05 time scale. The bold black and dashed red lines**
**show the modelled atmospheric and ocean mixed layer $\Delta^{14}$C curves after synchronization to the $^{14}$C records (yellow:**
**Lake Suigetsu; blue: Hulu Cave; purple and white: corals. The inset panel shows the PDF of the inferred timescale**
**difference between GICC05 and the combination of all $^{14}$C records. The black line is based on using only the**
**modelled atmospheric $\Delta^{14}$C. The red dashed line is based on comparing coral and speleothem data to the modelled**
**mixed-layer $\Delta^{14}$C, and Lake Suigetsu data to modelled atmospheric $\Delta^{14}$C. The green line shows modelled $\Delta^{14}$C based**
**on geomagnetic field changes.**




### 4.3 39,000 – 45,000 years BP

Our oldest tie-point is the previously discussed Laschamp event around 41 ka BP. The only independently and absolutely dated $^{14}$C record around this time that has a sufficient sampling resolution is the Bahamas speleothem by Hoffmann et al. (2010). While offset in absolute $\Delta^{14}$C (see Fig. 1), the U/Th-dated coral data supports the amplitude and timing of the $\Delta^{14}$C increase seen in the speleothem even though precise synchronization is hampered by the low sampling resolution of the corals. The Lake Suigetsu record is characterized by large uncertainties and scatter around this time. As discussed in section 3.3.2, IntCal13 is smoothed around Laschamp, having a smaller amplitude and a less sharp rise in $\Delta^{14}$C. For this tie-point, we merely remove the error-weighted mean between 39-45 ka BP from each dataset, since detrending would remove the largest part of the signal. Hence, there are large $\Delta^{14}$C modelling uncertainties associated with unknown carbon-cycle changes, and we assume a Gaussian $\pm1\sigma$ error of 50‰, which we consider conservative since sensitivity experiments imply that the impact of carbon cycle changes on $\Delta^{14}$C was likely below 40‰ (Köhler et al., 2006).

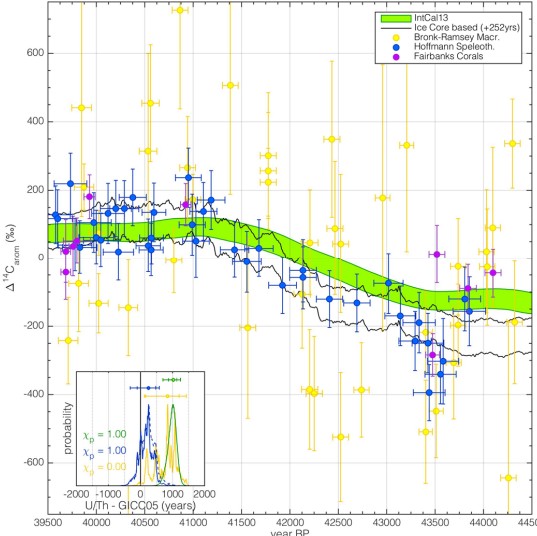

**Figure 10. Synchronization of $^{10}$Be and $^{14}$C around the Laschamp event. The black lines encompass the modelled $\Delta^{14}$C anomalies ($\pm1\sigma$) from the ice-core data shifted by +252 yrs (68.2% confidence interval = -103 to 477 yrs) according to their best fit to the speleothem $^{14}$C data. The green patch shows the $\pm1\sigma$ envelope of IntCal13. The blue and purple symbols show $\Delta^{14}$C from Bahamas speleothem, and corals, respectively. The yellow symbols show $\Delta^{14}$C anomalies based on Lake Suigetsu macrofossils. All datasets have been centred to 0‰ by subtracting the error-weighted mean of each dataset. The inset shows the PDF of the inferred age differences between the ice-core data and IntCal13 (green), Lake Suigetsu (yellow) and the Bahamas speleothem (blue). The dashed blue line corresponds to age differences from the modelled mixed layer $\Delta^{14}$C and the Bahamas speleothem.**

Synchronizing the ice-core stack to the speleothem, Lake Suigetsu, and IntCal13 data yields significantly different results. We infer that GICC05 produces ages about 250 years younger than the U/Th dated speleothem data (Fig. 10). The IntCal13 record however, implies a larger difference of ~1,000 years. Using Lake Suigetsu data, on the other hand, leads to multiple probability peaks of which two are in agreement with the speleothem, and one with the IntCal13 record. The large scatter of the Lake Suigetsu data however, leads to poor statistics (low $\chi^2$ probabilities). Furthermore, the Lake Suigetsu timescale is only constrained by varve counting back to




39 ka BP and based on extrapolation for older sections (Bronk Ramsey et al., 2012) and hence, provides less
precise constraints on the timing of the $\Delta^{14}$C increase.
To test which of these links is the most likely we turn to independent radiometric ages of the Laschamp
excursion. Pooled Ar-Ar, K-Ar, and U/Th ages on lava flows place the period of (nearly) reversed field direction
at 40,700 ± 950 yr BP (Singer et al., 2009), or 41,300 ± 600 yr BP (Laj et al., 2014). In addition, a North
American speleothem provides a U/Th-dated transient evolution of the geomagnetic field (Lascu et al., 2016),
with the lowest intensities occurring at 41,100 ± 350 yr BP. Comparing the ice-core $^{10}$Be stack to these data
clearly shows that all of these records rule out the +1,000 year time shift implied by IntCal13, as it would induce
a significant disagreement between radiometrically dated magnetic field records and the dating of the $^{10}$Be peak
in the ice cores (Fig. 11). We hence argue that the 252 yr offset inferred from the comparison to the Bahamas
speleothem is the most likely estimate of the timescale difference between GICC05 and the U/Th timescale
around this time. Similar as before, assuming that the speleothem represents a mixed-layer signal instead of
direct atmospheric $\Delta^{14}$C does not significantly affect the inferred timescale differences (see Fig. 11 inset, blue
dashed line).

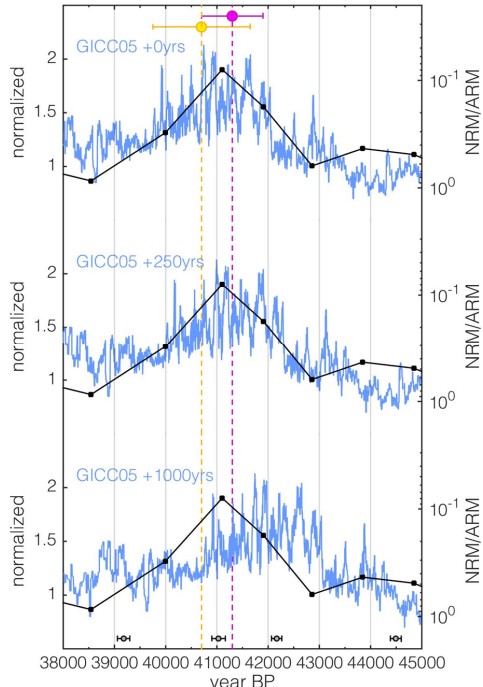


**Figure 11: Comparison of the ice-core stack (blue) to Ar-Ar dates of the Laschamp excursion (yellow: Singer et al. 2009, pink: Laj et al. 2014), and relative geomagnetic field intensity (black, NRM/ARM, reversed y-axis) from a U/Th-dated speleothem (Lascu et al., 2016). The individual speleothem U/Th dates are shown on the bottom of the figure with their ±2σ uncertainties. Each panel shows a different shift of GICC05 according to the results from figure 10.**

546



### 4.4 Transfer Function

To construct a continuous transfer function between GICC05 and the U/Th timescale we apply a Monte-Carlo approach. We randomly sample the PDFs of the timescale differences for each tie-point established in the previous sections (Fig. 7, 9, 10). For the interpolation in-between and the corresponding uncertainties we consult the GICC05 maximum counting error (mce). Since the layer counting uncertainty is incremental, we use the time derivative of the mce to estimate interpolation uncertainty. We generate AR(1) noise ($\mu = 0$, $\sigma = 1$), multiply it with the derivative of the mce, and calculate the cumulative sum back in time. As shown during the Holocene (Adolphi and Muscheler, 2016), the mce is not a random uncorrelated Gaussian error, but appears to be systematic. Hence, we use an AR-process with a strong autocorrelation. We note that this treatment of the mce leads to larger interpolation errors compared to assuming a white noise model, which would lead to very small uncertainties that average out over long time (see also discussion in Rasmussen et al., 2006). Furthermore, we treat the mce as $\pm 1\sigma$ instead of $\pm 2\sigma$ as proposed by Andersen et al. (2006) which additionally increases our interpolation error. We note that this procedure does not aim to provide a realistic model of the ice-core layer-counting process and its uncertainty which is clearly more complex (see Andersen et al., 2006; Rasmussen et al., 2006), nor should it be interpreted such that the mce was a $1\sigma$ uncertainty. However, our approach allows us to infer a conservative estimate of the interpolation uncertainty while at the same time it takes advantage of the fact that GICC05 is a layer counted timescale and hence, cannot be stretched/compressed outside realistic bounds. The resulting realizations of the AR(1) process are then used to interpolate between the tie-points established from sampling the PDFs of the $^{14}$C/$^{10}$Be matches. This procedure was repeated 300,000 times which was found sufficient to obtain a stationary solution, leading to 300,000 possible timescale transfer functions.

Figure 12 shows the resulting mean transfer function along with its confidence intervals. Firstly, it can be seen that all tie-points fall into the uncertainty envelope of GICC05. The implied change in the timescale difference between the youngest two tie-points (i.e., over the course of GS-1), and between 13,000 and 22,000 years BP is slightly larger than allowed by the mce, albeit the latter is consistent within the uncertainties of the tie-point at 22,000 years BP. We can see that the use of the mce to determine the interpolation error leads to small uncertainties wherever the change in the timescale difference is large (e.g. over the 13,000 – 22,000 years BP interval): Stretching GICC05 by as much as the counting error allows, requires that every uncertain layer has in fact been a real annual layer, leaving little room for additional error. Between 22,000 and 42,000 years BP, the interpolation uncertainties are determined by the mce and thus, grow/shrink at a rate determined by the mce.

Our results are in very good agreement with the results by Turney et al. (2016) around Heinrich 3. In this study, a kauri-tree $^{14}$C sequence was calibrated onto Lake Suigetsu $^{14}$C and also matched on GICC05 via $^{10}$Be. The difference of the inferred ages (i.e., kauri on Suigetsu vs. Kauri on GICC05) matches with our proposed transfer function (red star in Fig. 12).

Figure 12 also shows the inferred offset between the $^{40}$Ar/$^{39}$Ar-age of the Campanian Ignimbrite (Giaccio et al., 2017) and a tentatively attributed SO$_4$-spike in the GISP2 ice core (Fedele et al., 2007). Even though it obviously requires a well-characterized tephra find in the ice cores to ensure that the SO$_4$-peak is indeed associated with the Campanian Ignimbrite, at least from a chronological point of view, our transfer function does not preclude this link. However, no matching shards were identified in this period (Bourne et al., 2013).





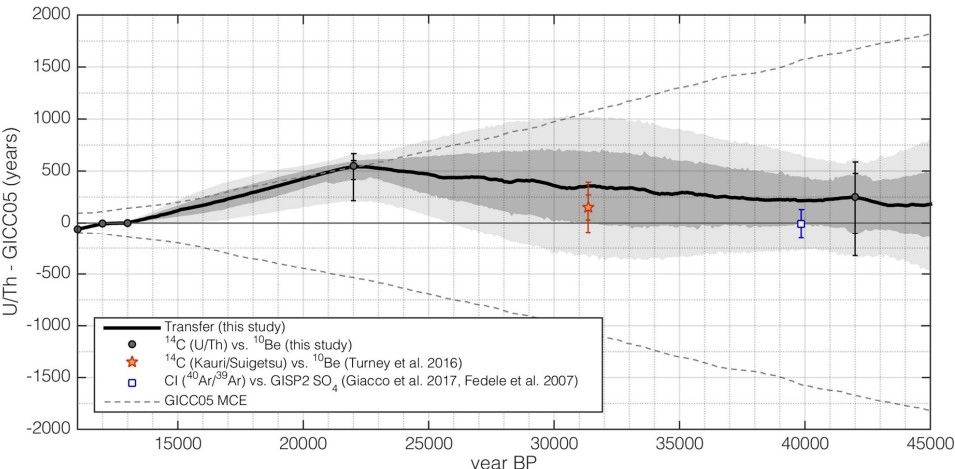

**Figure 12: Transfer function between the U/Th timescale and GICC05. The transfer function is shown in black with dark and light grey shading encompassing its 68.2% and 95.4% confidence intervals. The black dots with error bars show the used match points between $^{14}$C and $^{10}$Be. The red star shows the difference between ages of a glacial kauri tree $^{14}$C sequence on Lake Suigetsu $^{14}$C and GRIP $^{10}$Be (Turney et al., 2016). The blue open square shows the age difference between the $^{40}$Ar/$^{39}$Ar-age of the Campanian Ignimbrite (Giaccio et al., 2017), and a tentatively associated spike in the GISP2 SO$_4$ record (Fedele et al., 2007) on the GICC05 timescale (Seierstad et al., 2014).**

## 5 The timing of DO-events

To investigate the synchronicity of climate changes recorded in different parts of the globe, we compare ice-core data to a selection of well-dated speleothem records. The well-known Hulu-Dongge Cave records have become iconic blueprints for intensity changes of the East Asian Summer Monsoon (EASM) anchored on a precise U/Th timescale (Cheng et al., 2016; Dykoski et al., 2005; Wang et al., 2001). The speleothem records from Cueva del Diamante and El Condor reflect changes in precipitation amount over eastern Amazonia associated with the South American Monsoon (Cheng et al., 2013b). The speleothem records from Sofular Cave, Turkey, are not straightforward in their mechanistic interpretation but likely reflect a mix of temperature and seasonality of precipitation ($\delta^{18}$O), and type and density of vegetation, soil microbial activity ($\delta^{13}$C), and hence, effective moisture and temperature (Fleitmann et al., 2009). Hence, while this list of speleothem data can certainly be expanded in future studies, we chose these four speleothem records from 3 different regions that are all well-dated and sensitive to the position of the ITCZ and compare it to the ice-core records.





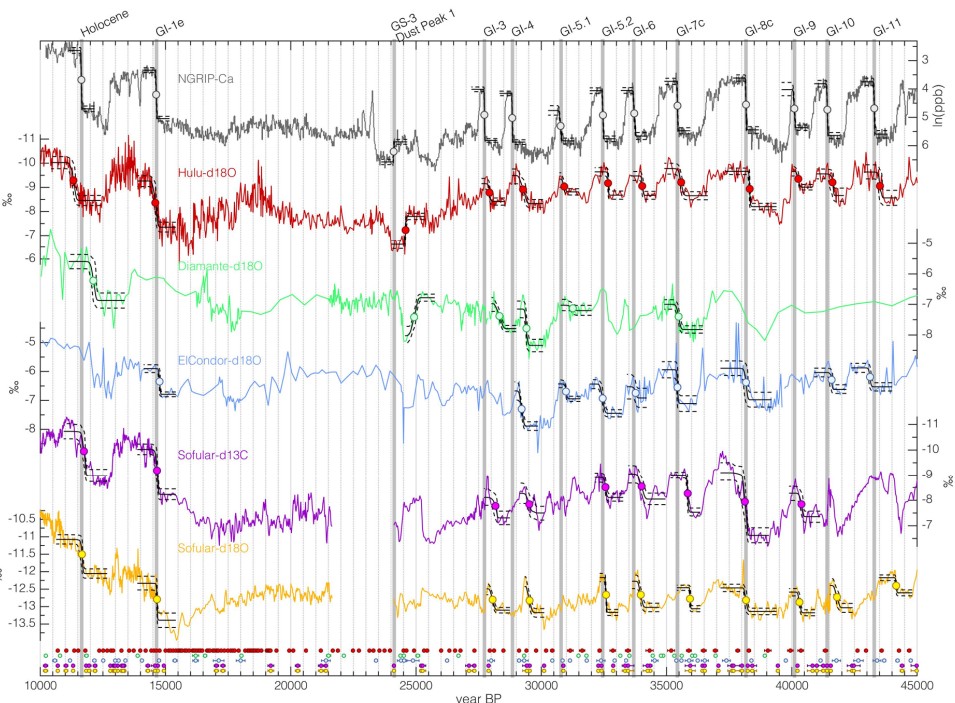

**Figure 13: Timing of abrupt climate changes in different climate records. The climate archive and proxy is indicated in each panel. The black lines show the mean of the fitted ramps and their 95% confidence intervals (dashed). The dots mark the midpoint of the mean transition. The U/Th dates and their ±1σ uncertainties of each climate record are shown at the bottom of the figure in colour coding corresponding to the respective climate record. Each time series is shown on its original timescale not applying any synchronization.**

Figure 13 shows the ice-core and speleothem climate records on their original individual timescales, along with the fitted ramps to the rapid climate changes. Note that we could not fit each climate event for every record, since the method requires a minimum number of data points defining the levels before and after each transition to produce reliable estimates. Already visually, a lag of climate changes in Greenland compared to the speleothem records can be consistently identified between 20 and 35 ka BP when all records are on their original timescales. Combining the PDFs of the detected change points in Greenland and the speleothems allows us to infer a probability estimate of the timing difference between climate events in Greenland and speleothems. These differences are shown in figure 14 along with our transfer function based on the matching of radionuclides from figure 12. This comparison shows that the differences in the timing of start-, mid- and end-point of DO-events in speleothems and ice cores largely fall within the uncertainties of our radionuclide-based timescale transfer function. Thus, rapid climate changes occur synchronously in Greenland and the (sub-) tropics. Notable exceptions are i) the transition from GS-1 to the Holocene around 11.6 ka BP, ii) Heinrich event 2 at 24 ka BP, and iii) DO-11 around 43 ka BP. However, there is large scatter among the different speleothem-based estimates at these events, indicating that these events are asynchronous in the different speleothems records on their respective timescales. Consequently, some of these records also imply asynchronous climate shifts with Greenland ice cores. This may either be interpreted as an indication of time-transgressive climate





changes, or as a bias in individual speleothems – either in how climate is recorded in the speleothem, or their

dating (for example through detrital thorium).

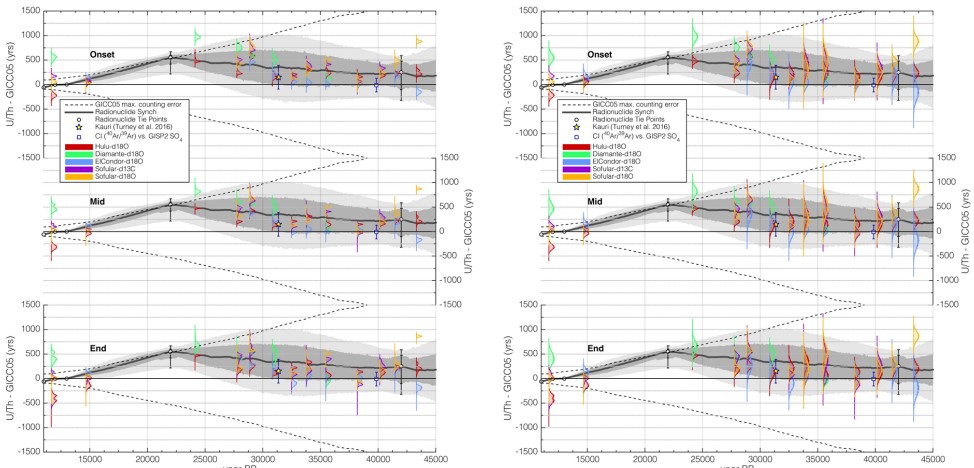

**Figure 14: Timing differences of the onset (top), midpoint (middle) and end (bottom) of rapid climate changes in NGRIP and speleothems (coloured PDFs, see legend), and the timescale transfer function inferred from radionuclide matching (black line and grey shadings as in figure 12). The left panels show the PDFs of timing differences including only uncertainties from the determination of the change points in the climate records, while the right hand panels also include the speleothem dating uncertainties.**

Averaging over all DO events, we can estimate an overall probability of leads and lags. By using the individual

realizations of the radionuclide-based transfer function (see section 4.4) we take into account that the

uncertainties of the transfer function are strongly autocorrelated. For each realization, we randomly sample the

PDFs for the onset of the DO-events for the ice-core and speleothem records, perturb the speleothem-based

estimates within their U/Th dating errors, determine the lead or lag between the DO-onset in ice-core and

speleothem records, and correct it for the expected lag from the realization of our transfer function. By

averaging over all DO-events we thus obtain a mean lag for each realization and speleothem. In addition, we

combine the different speleothem-based estimates of each realization by averaging over their mean lags to

obtain an overall (speleothem & DO-event) mean lag. Converting the obtained lags from each realization into

histograms we estimate the PDFs of average lags between ice-core and speleothem records.

Our lag estimates critically depend on our ability to fill the gaps between the widely spaced tie-points and thus,

on our assumptions about the ice-core layer counting uncertainty, and how well our AR(1) process model can

capture these (section 4). However, we note that by treating the mce as a highly correlated $1\sigma$ (instead of $2\sigma$)

uncertainty, our error estimate can be regarded as very conservative since it allows for large systematic drifts in

each realization of the transfer function that will result in large errors of the mean.

The resulting PDFs of the lag between speleothems and ice cores are shown in figure 15. The uncertainties are

mainly determined by our synchronization uncertainty. Thus, the uncertainty is only marginally reduced when

averaging over all speleothems (Fig. 15, bottom): Because each realization of the transfer function varies

smoothly, the offset between speleothem and ice-core records will be systematic for all speleothems in each

realization, and is thus only marginally reduced by averaging.




656   We find that all speleothem records except Cueva del Diamante (Cheng et al., 2013b) indicate synchronicity

657   with NGRIP within 1σ and that the delay obtained for Cueva del Diamante falls within 2σ. We note that the

658   speleothem data from El Condor (Cheng et al., 2013b) from the same region as Cueva del Diamante does not

659   indicate a significant lag to Greenland. Overall, our analysis cannot reject the null-hypothesis of synchronous

660   DO-events in Greenland ice cores and (sub-) tropical speleothems (lag: μ±1σ = 29±189 years).

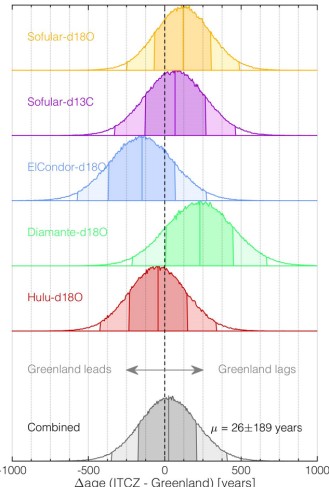

661

**Figure 15: Average lead/lag between the onset of DO-events in the speleothems and NGRIP. Each panel (colour)**
**shows the PDF for the lead/lag of the onset in the speleothem compared to NGRIP, averaged over all investigated**
**DO-events (i.e., excluding the GS-3 Dust Peak/H2). The bottom most panel shows the PDF of the average of all DO-**
**events and speleothems. The dark/light shading of the PDF in each panel indicates 68.2%/95.4% intervals.**

**6 Discussion**
Our proposed transfer function quantifies the long-term differences between the Greenland ice-core and U/Th
timescale and allows their synchronization. Even though based on only a few tie-points, this can be used to
evaluate the absolute dating accuracy of Greenland ice-core records during the past 45 ka BP, while maintaining
the strength of their precise relative dating. In combination with similar work done for the Holocene (Adolphi
and Muscheler, 2016; Muscheler et al., 2014a), the picture emerges that the GICC05 counting error may be
systematic: when accumulation and data resolution is high (e.g. in parts of the Holocene), too many annual
layers have been counted, whereas during periods of low accumulation (e.g. GS-1 and GS-2) there is a tendency
to identify too few annual layers. In principle, this is well captured by the GICC05 uncertainty estimate as the
derivative of our transfer function is (within error) consistent with the increase of the counting error. However,
our results caution against the use of the GICC05 counting error as a 2σ uncertainty as is often done in the
literature. Originally, Andersen et al. (2006) pointed out that the MCE is not a true σ uncertainty but proposed
that a Gaussian distribution with 2σ = MCE could serve as a pragmatic approximation. In combination with
results from the Holocene (Adolphi and Muscheler, 2016) our study shows that the counting error can be
strongly correlated over extended periods of time. This is in line with the discussion in Rasmussen et al. (2006)
who point out that the main contribution to a potential bias in the layer count is the definition of how an annual
layer is manifested in the proxy data. The data resolution as well as the manifestation of annual layers change



between different climate states (Rasmussen et al., 2006), likely due to changes in aerosol transport and
deposition resulting from variations in the atmospheric circulation and seasonality of precipitation (Merz et al.,
2013; Werner et al., 2001). According to our analysis, the largest relative (i.e., year/year) change in the
difference between GICC05 and the U/Th and tree-ring timescale occurs over GS-1 (11,653-12,846 years BP)
and GS-2 (14,652-23,290 years BP). Both of these periods have likely been characterized by an increased
relative contribution of summer precipitation to the annual ice layer (Werner et al., 2000; Denton et al., 2005),
and the annual layers in the ice core have been identified in a similar way in both intervals (Rasmussen et al.,
2006). In the 11-13 ka BP interval, the offset between GICC05 and the tree-ring timescale changes from -60
(95.4%-range: -77 to -42) years to zero (95.4%-range: -12 to +21) years. During the same interval, the GICC05
maximum counting error grows by 46 years. Albeit consistent within the absolute error margins, this stretch of
GICC05 over GS-1 thus slightly exceeds the range allowed by the GICC05 counting error. Muscheler et al.
(2014a) discussed that this stretch may be partly explained by errors in the placement of the oldest part of the
tree-ring chronologies. However, here, we use a revised late glacial tree-ring dataset in which the different
chronologies are connected much more robustly (Hogg et al., 2016). Furthermore, our analysis on the fully
independent Hulu Cave $^{14}$C data yields similar results (Fig. 7). Hence, our analyses clearly show that the GS-1
interval is about 60 years too short in the GICC05-timescale.
Between 15 and 22 ka BP, our analysis yields a change in the GICC05 offset from +118 (95.4%-range: 2-220)
years to +549 (95.4%-range: 207-670) years, while the GICC05 counting error grows by 335 years. Thus, again,
our transfer function changes a little faster than the maximum counting error allows during this interval. We
note that our $^{14}$C-$^{10}$Be matchpoint around 22,000 years BP has a relatively low signal-to-noise ratio in the $^{14}$C
data (see Fig. 8-9) and should, thus, be regarded as tentative. However, as shown in figure 8 our results are
generally robust against different choices of subsets of the $^{14}$C data and time windows. Nevertheless, it can also
be seen that the estimates of the most likely age difference (i.e., the peak of the PDFs) differ slightly for
different choices of the $^{14}$C data. Hulu Cave yields a most likely offset of ~325 years, while Suigetsu implies a
bigger age difference of ~550 years that coincides with a secondary probability peak in the Hulu Cave PDF. We
note that assuming increased amounts of old soil organic carbon contributing to the speleothem formation would
lead to an even stronger difference between these estimates (see section 3.4). Hence, we propose an age
difference of +549 (95.4% range: 207-670) years based on the combination of all data (Fig. 9) that is consistent
within error with the estimates based on the single datasets shown in figure 8, but stress that this tie-point should
be re-evaluated as new suitable $^{14}$C data becomes available in the future.
Assuming that the U/Th dates are absolute, our transfer function can be used to account for the bias in the
GICC05 timescale and thus facilitate comparisons of ice-core records to other absolutely dated archives.
However, we note that our synchronization does not necessarily lead to consistent timescales with radiocarbon-
dated records. As discussed in section 3.3.2 (Fig. 4) and section 4.3 (Fig. 10 & 11), discrepancies of the datasets
underlying IntCal13 can lead to erroneous structures in the calibration curve. The reduced amplitude of the $\Delta^{14}$C
change around the Laschamp geomagnetic field minimum in IntCal13 compared to its underlying data implies
that IntCal $\Delta^{14}$C must be offset prior to and/or after the Laschamp event. This underlines the challenges in
radiocarbon calibration around this time pointed out by Muscheler et al. (2014b). Also more recently, Giaccio et
al. (2017) pointed out that paired $^{40}$Ar/$^{39}$Ar and $^{14}$C-dating of the Campanian Ignimbrite around 40 ka BP yields
inconsistent ages when the $^{14}$C age is calibrated with IntCal13. Since IntCal13 in principle should be tied to the



U/Th-age scale, this implies either an inconsistency between Ar/Ar and U/Th dating or in the reconstructed $^{14}$C
levels of the calibration curve. The latter would be congruent with the conclusions by Muscheler et al. (2014b).
If the problem was indeed the IntCal $^{14}$C reconstruction, a synchronization of ice-core $^{10}$Be to IntCal $^{14}$C would
not resolve this bias, since the problem would not be one of chronology, but of $^{14}$C measurement and/or archive.
Our analysis provides the first rigorous test of whether DO-events recorded in speleothems and ice cores occur
synchronously. We find that on average, the onset of DO-events occurs simultaneously within the precision of
our method ($\pm 1\sigma = \pm 189$ years), consistent with the findings of Baumgartner et al. (2014). Since we compare to
speleothem records from different regions, this also highlights that the ITCZ likely migrated synchronously
(within uncertainties) over the different ocean basins and continents during the onset of DO-events. However,
there are also differences between the different speleothem records, which could be due to limitations in their
dating or related to how directly individual archives record the rapid climate changes. The most notable
examples are the onset of the Holocene and GI-11, which appear to occur asynchronously in the different
speleothems (see Fig. 13 & 14). Another example is the younger GS-3 dust peak in the Greenland ice cores that
appears to coincide with the East Asian Summer Monsoon decline seen in Hulu Cave, but postdates the
precipitation increase seen in El Condor and Diamante. This change in the speleothems is typically attributed to
the southward shift of the ITCZ as a response to Heinrich Event 2.
Figure 16 shows the period around H2. Firstly, we note that the younger of the two GS-3 dust peaks in the
Greenland ice cores (Rasmussen et al., 2014a) occurs coevally (within chronological uncertainty) with the ITCZ
movement recorded by the speleothems. At this time, the East Asian Summer Monsoon is strongly reduced as
implied by decreased Hulu Cave $\delta^{18}$O (Cheng et al., 2016). Coevally, precipitation increases in the South
American Summer Monsoon region (Novello et al., 2017; Stríkis et al., 2018). The records thus exhibit more
pronounced stadial conditions than normally seen during (non-Heinrich) DO-events. However, taken at face
value, the precipitation increase at El Condor and Cueva del Diamante, the two northernmost sites shown in
figure 16 (Cheng et al., 2013b), significantly predates the event seen in Greenland and Hulu Cave. It also
predates the more southern South American sites Lapa Sem Fim (Stríkis et al., 2018) and Jaragua (Novello et
al., 2017) by more than 500 years. This could either point to errors in the dating of the El Condor and Diamante
speleothems, or be related to their latitudinal position. A freshwater-only experiment (all other boundary
conditions held constant at 19 ka BP levels) with the Community Climate System Model 3 (TraCE-MWF, He,
2011) shows that, during a weak AMOC state, reduced advection of moisture from the tropical Atlantic leads to
lower precipitation north of the ITCZ (Fig. 16). El Condor and Cueva del Diamante are both located very close
to the LGM position of the ITCZ. It is hence possible, that when northern hemisphere summer insolation
reached its lowest values over the past 50 kaBP around H2, the ITCZ migrated to a position south of El Condor
and Cueva del Diamante. As a result, the precipitation response to freshwater forcing would change sign at these
cave sites. The sites located slightly further south only show a weak (Pacupahuain) or no (Paixao) response
during this period, but are both characterized by increased variability. The two southernmost sites on the other
hand (Jaragua and Lapa Sem Fim) remain south of the ITCZ throughout, and hence, show a clear increase in
precipitation coeval with the signal in Greenland and Hulu Cave. In this context, the precipitation increase in El
Condor and Cueva del Diamante around 25kaBP (i.e., prior to H2) may signify when the ITCZ transitions over
the sites. The subsequent reduction in AMOC strength during H2 then leads to a decrease in precipitation in
north-west South America, an increase further south, and little change in between. Tentative support for this can





be drawn from the response of the El Condor and Cueva del Diamante speleothems to GI-2.2 and GI-2.1 where,
albeit weakly, the $\delta^{18}$O records imply an increase in precipitation during GI-2 which is opposite to their
response to DO-events during MIS-3 (Fig. 13, 16). Thus, this analysis indicates, that the seemingly
asynchronous response to climate change in different proxy records may indeed only reflect site specific
changes in the proxy response. Alternatively, we cannot rule out undetected issues with the U/Th ages of these
speleothems. A detailed analysis of this observation feature is beyond the scope of this paper, but in the context
of a timescale perspective, which is the focus of this work, it highlights the caveats of climate wiggle-matching
between single records, even if the mechanistic link between regional climate changes may be relatively well
understood.

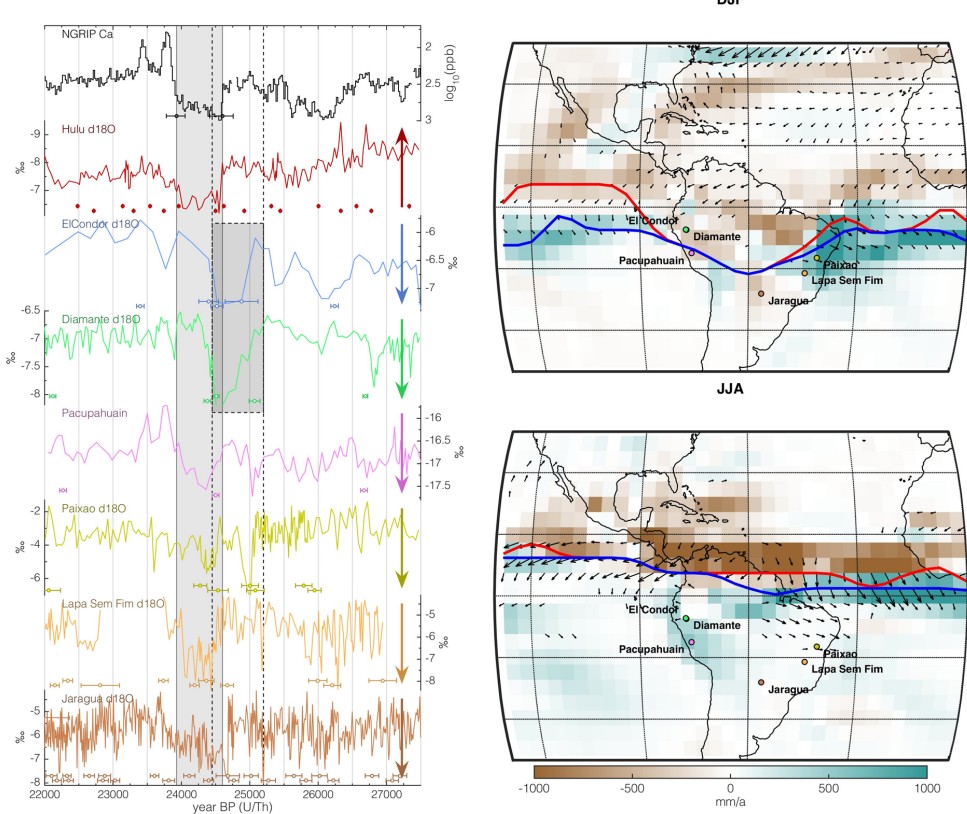

**Figure 16: Climate changes around H2. Left (from top to bottom): NGRIP Ca (Bigler, 2004) on the synchronized timescale (Fig. 14), Hulu Cave $\delta^{18}$O (Cheng et al., 2016), El Condor $\delta^{18}$O (Cheng et al., 2013b), Cueva del Diamante $\delta^{18}$O (Cheng et al., 2013b), Pacupahuain $\delta^{18}$O (Kanner et al., 2012), Paixao $\delta^{18}$O (Stríkis et al., 2018), Lapa Sem Fim $\delta^{18}$O (Stríkis et al., 2018), Jaragua Cave $\delta^{18}$O (Novello et al., 2017). The arrows on the right hand side of each axis point in the direction of the signature of increased precipitation on $\delta^{18}$O through the amount effect (Dansgaard, 1964). The light grey box marks H2. The dark grey box highlights the preceding $\delta^{18}$O anomaly in El Condor and Diamante caves. Right: Precipitation (colour) and wind (arrows) response to freshwater forcing in the CCSM3 climate model (freshwater only experiment of TraCE21k, all other forcings are held at 19k conditions, He, 2011). The red (blue) line depicts the latitude of the precipitation maximum during strong (weak) AMOC-states. Only wind anomalies >1m/s are plotted. The cave sites are indicated as dots. The top panel shows the winter (December-February) response, while the bottom panel shows the summer (June-August) response. Anomalies are plotted as weak-strong AMOC mode.**




## 7 Conclusion

We present the first radionuclide-based comparison between the Greenland Ice Core Chronology 2005 (GICC05) and the U/Th timescale. We find that GICC05 is accurate within its stated absolute uncertainties, but also that the maximum counting error of the GICC05 may be at the limit to capture the total uncertainty accumulated within certain climatic periods. Our analysis indicates that the relationship between GICC05 and the U/Th timescale over the last 45 ka drifts over time and reaches its maximum offset around 22 ka BP. We propose a transfer function that quantifies this drift and facilitates analysis of ice-core and U/Th records, such as speleothems, on a common time scale. Thus, this transfer function allows further integration of key-timescales in paleosciences and contributes to the INTIMATE (INTegration of Ice-core, MArine, and TErrestrial records) initiative (Bjorck et al., 1996; Rasmussen et al., 2014b; Bronk Ramsey et al., 2014). Provided that U/Th ages are regarded accurate, the transfer function reduces the absolute dating uncertainty of Greenland ice cores by 50-70% back to 45 ka BP. We find that, on average, the onset of DO-events occurs synchronously in Greenland, East Asia, and South America. We show that the southward ITCZ shift around 24.5 ka BP seen in speleothems, typically associated with H2, coincides with the younger GS-3 dust peak recorded in Greenland ice cores. However, we also highlight inconsistencies between speleothem records around the onset of the Holocene, late GS-3, and GI-11 and thus, caveats to the commonly applied practice of climate wiggle-matching.

By comparing various [14]C records underlying IntCal13 as well as ice-core [10]Be data and geomagnetic field records, we infer that the current radiocarbon calibration curve underestimates the amplitude and rapidity of the $\Delta^{14}$C change around the Laschamp event 41 ka BP. This adds to previous studies (Giaccio et al., 2017; Muscheler et al., 2014b) highlighting that there are likely systematic errors in IntCal13 that will directly translate into errors of radiocarbon-based chronologies around that time. The combination of several internally inconsistent datasets in IntCal13 can lead to erroneous timing and amplitude of $\Delta^{14}$C changes. Hence, great care has to be taken when attempting to use sections older than 30 ka BP of IntCal13 directly for studies of [14]C production rates and/or carbon cycle changes.

## 8 Data Availability

The transfer function shown in figure 12 will be made available as a supplementary to this paper and on NOAA.

## 9 Author contributions

FA designed and carried out analyses, and wrote the manuscript in correspondence with CBR and RM. TE designed and applied break-point detection analysis and wrote the corresponding methods section. FA, RM, CBR, SOR, CT and AC initiated the project. RLE and HC provided speleothem data. AS and SOR provided insights into the ice core chronology. HF and TE gave insights into aerosol transport and deposition. All authors discussed and commented on the manuscript.



## 10 Competing interests

The authors declare that they have no conflict of interest.

## 11 Acknowledgements

FA was supported through a grant by the Swedish Research Council to FA (Vetenskapsrådet DNr. 2016-00218). CBR was partially supported through the UK Natural Environment Research Council (NERC) Radiocarbon Facility (NRCF010002). TE and HF acknowledge the long-term support of ice-core research at the University of Bern by the Swiss National Science Foundation (SNSF) and the Oeschger Center for Climate Change Research. SOR gratefully acknowledges support from the Carlsberg Foundation to the project ChronoClimate. This work was partially supported by the Swedish Research Council (grant DNR2013-8421 to RM), the NSF 1702816 to RLE, and the Australian Research Council DP170104665 to CT and AC. We gratefully acknowledge the financial support of the University of Adelaide Environment Institute, for the initial Marble Hill meeting that initiated this work.

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
