# Peer review of "Connecting the Greenland ice-core and U/Th timescales"

_Climate of the Past, 2018_

## Referee Comment (RC1) · N. Boers (Referee) · 8 Aug 2018

Summary:

This paper provides a very thorough synchronization of the GICC05 time scale obtained from counting annual layers in ice cores, and (assumed to be) absolute U/Th dates from several (sub-)tropical speleothems via cosmogenic radionuclides with a focus on 14C, for the time period from 10ka to 45ka BP. Based on this synchronization, the timing of the DO events during this interval is compared among ice core and

speleothem records, and it is concluded that on average, no systematic lead or lag can be inferred, given the inherent uncertainties.

The paper is written very well, the subject is of great scientific importance, and the employed methodology seems accurate to me. I hence strongly support publication of this study in CP.

However, there are some instances where the presentation is not detailed enough at least for me to be able to precisely follow what is done exactly (see specific comments below). In addition, I have some slight conceptual concerns that I would suggest to be addressed prior to publication. Please note that I'm not a geochemist, so I apologize in advance for potentially trivial or irrelevant questions / concerns below.

Major comments:

1. Necessarily, some of the uncertainties are put in 'by hand', such as treating the MCE of the GICC05 time scale as 1 sigma, but also at several instances of the analysis of the cosmogenic radionuclides. This is not a critique per se, and I agree with the authors that their uncertainty estimates are probably very conservative. However, in the situations at hand, it cannot be quantified _how_ conservative, and this leads to a tricky situation: the more conservative the error estimates are chosen along the way, the harder it is to reject the null hypothesis of synchronous DO events in the different records. The final sigma reported for the average over all DOs and speleothems is 189yr, and a lot can happen with such uncertainties; the statement in the conclusion that on average the DO onsets occurred synchronously is thus misleading, I think. I'd suggest to rather emphasize here that no systematic leads or lags can be inferred given the (partly subjectively introduced) error estimates. In addition, the 189yr are not far from the delay between NGRIP and WAIS inferred to be significant by the WAIS members, would you mind to comment on this?

2. It is stated in the abstract , introduction, and in the discussion that the GICC05 uncertainties are reduced by 50-70%, but I don't understand where these numbers

come from, and as far as I can tell, they are not mentioned / explained somewhere else in the manuscript. If you compare the GICC05 MCE to the sigma of the transfer function ensemble, it might be problematic, since the MCE is not really related to a normal distribution, despite the pragmatic approach by Andersen et al.

3. It is not clear to me how exactly the interpolation in Sec. 4.4 is carried out. This is a key part of the study, and I would hence suggest to make this section considerably more detailed. It is written that the AR(1) realizations are used for interpolation, but how? You sample from the PDFs at the tie points, but how do you make sure that a given AR(1) realization, starting at one tie point, ends up close to the next tie point? Note that I might be completely off track here.

Specific comments:

p3, l83: How were the 50-70% uncertainty reduction inferred quantitatively?

p3, l91: the "Hence" suggests that the previous sentence implies the _inverse_ relationship, but I don't think it does, although I don't question the inverse relationship itself.

p6, l179: what do you mean by "more direct function of the timescale?"

p6, l190ff: Using both flux-corrected and non-corrected version of the ice core records is fine to infer systematic differences between the records via comparison to the expected error of the mean, but I find it a bit problematic to use such a stack for the synchronization; do you obtain different results when using only flux-corrected or only non-corrected versions of the records?

p7 l215: please define "cal"

p7 l221: Could you motivate the assumption of proportional 10Be and 14C production rate changes here?

Fig.2: the time scales are not equidistant, right? How do you perform the the FFT

filtering? Do you interpolate? If yes, using which method, and to which resolution?

p10, l315: It may be my fault, but where in the results section are the window length and frequencies given? Can you be more specific?

p10, l324: I don't understand this sentence: do you mean that the delay between associated peaks in different sinusoidal signals increases with wavelength? Why?

p11, l328: is the box-diffusion model by Siegenthaler et al referred to here?

p13, l284f: what do you mean by "deviations from the transition"? I find it a bit problematic to refer the reader to a paper in preparation here, since it's not clear given the presentation here, how the change-point detection is carried out. In particular, further below it becomes clear that for each potential change point, PDFs are obtained for its onset, mid point, and end point, but it's not clear how these PDFs are derived.

Fig.7: -there seems to remain quite a discrepancy between the variability of the bold grey line and the Towai treering data (green) even after synchronization, could you comment on this? - if I'm not mistaken, none of the time scales of the shown data are equidistant, how to you do the FFT filtering in this case? If you interpolate, how?

p15, l445: can you explain what you mean by "to remove offsets"? This also relates to l312 on p10; are't offsets at longer time scales potentially problematic? I guess _heuristically_ these longer-term offsets are attributed to carbon cycle changes, but a clarifying sentence would be good, I think.

Fig.8: You present the result of the synchronization, and show the PDFs for the different windows, but I think one or two extra sentences in Sec 3.4 on how exactly these PDFs are used to shift the record across the windows would be very beneficial.

Fig11: there's no inset and no blue dashed line! Also, shouldn't the four individual speleothem dates correspond to the measured (black) points of NRM/ARM?

p.19: As noted above, I don't understand how you interpolate between the tie points,

the description is too brief in my opinion: by derivative of the MCE, do you mean the increments from one measured point to the next? Why do you multiply the AR(1) with these? Which "cumulative sum back in in time", i.e. from where to where? You say "strong autocorrelation", but what is the value of the parameter alpha?

p19, l 575: what do mean by "grow/shrink at a rate determined by the mce"? the latter is cumulative and hence always increasing back in time, but your AR(1) based uncertainties decrease when going back in time towards the next time point. I agree that it should decrease this way, but I don't understand the method sufficiently from the given description to understand how, specifically.

p22, l639: Here you say that you sample the PDFs for the DO onsets; am I correct in assuming that for each onset, you obtain a PDF of its dates from the change-point-detection?

p23, l679: I don't think that this study _shows_ that the counting error can be strongly correlated over extended period; please correct me if I'm wrong!

p25, l727-738: it would be good, I think, to add reference on the relation to the ITCZ position already here.

p25, l739ff: The fact that the precip increase in El Condor and Cueva del Diamante significantly predates the onset of H2 in Greenland suggests that the southward shift of the ITCZ, proposed to explain the precip increase, was not caused by H2, but rather by long-term solar insolation changes and in particular the NH minimum around this time, right? Also, Fig.16 suggests that the variability in AMOC strength (related to H2) does not substantially affect the position of the ITCZ, but merely the precipitation anomalies north and south of the ITCZ. If this is correct, please revise the paragraph accordingly.

p27, l783: see above regarding the 50-70%

p27, l784: note the above comment on the formulation that DO events occur on average

synchronously, rather, the null hypothesis of synchronicity cannot be rejected given the uncertainties. Your statement in the abstract is more accurate, I think.

Sorry for the lengthy report, I hope it's helpful!

Best,

Niklas Boers

---

## Referee Comment (RC2) · J. Severinghaus (Referee) · 8 Aug 2018

This paper addresses a crucial problem we face in paleoclimatology - namely that many of us are going ahead and using the U-Th-dated speleothems to improve other paleo chronologies, without really having answered the fundamental question of whether the abrupt DO events seen in speleothems are synchronous with those seen in Greenland ice cores. I am as guilty of this as anyone - in Buizert et al. (2015) Clim. Past, 11, 153–173 we made a physical argument based on known atmospheric and oceanic processes that the Chinese speleothem DO events cannot have lagged Greenland's

[Figure]

DO events by more that 50 years. We then proceeded to tie the Greenland and WAIS Divide timescales in a pragmatic fashion to the Chinese speleothems, adopting an uncertainty of 50 years due to the assumption of synchroneity. I do believe that this argument is solid, but it is not enough for the high scientific standards we as a community must ultimately achieve, and the authors of the present paper are attempting to rectify this problem and empirically show that this lag cannot be very large. Therefore this work is essential, timely, and critical to the paleo field, and therefore I think this paper should be published with only very minor revisions.

The ultimate uncertainty that the authors arrive at is large, unfortunately, so it is perhaps best if the language of the conclusions is adjusted to reflect that large uncertainty. Instead of saying that the speleothems and ice cores are synchronous within uncertainty (which is true), it might be more helpful to the reader to write "we can reject the hypothesis of asynchrony larger than 189 yrs" or something equivalent. That way the conclusion shows what has actually been added by the present work.

Minor comments:

The term "synchronicity" is used in psychology (i.e. Carl Jung) and has nothing to do with paleoclimate or chronology. The proper term is "synchroneity". Please change all the uses in the paper accordingly.

---

## Editor Comment (EC1) · D.-D. Rousseau (Editor) · 8 Aug 2018

Dear authors,

Reviewer #1 N Boers did post his detailed review and I strongly encourage you to take advantage of this discussion phase to reply already his main comments so that you could engage some discussion that could help you later to prepare the potential revised when all the reviews will be completed. Thanks in advance,

All the very best. denis-didier Rousseau

---

## Referee Comment (RC3) · P. Reimer (Referee) · 10 Aug 2018

This manuscript uses cosmogenic isotopes to synchronize the Greenland ice core timescale with the U-Th timescale through a meticulous, multi-step process. The authors minimize the root mean square error in the production rate models from geomagnetic field based reconstructions and the ice cores to resolve the scaling factor for 10Be. They then compare 14C archives from around the Lachamps event with the reconstruction from the scaled ice core stack to select the most suitable ocean ventilation rate for the carbon cycle. The investigation into the effect of delay between ice

core reconstructed atmospheric 14C changes and the marine and speleothem archives was insightful. Once the ice cores were synchronized to the U-Th (and dendrochronological) timescale the synchroneity of the proxy response to D-O cycles in a number of speleothem climate records was tested. This represents a very important step in interpretation of palaeoclimate records. The ice core based 14C reconstruction will also provide a guide to improvements for the next IntCal radiocarbon calibration curve update.

Specific comments: p. 2, line 52-54 'About one third of the data underlying the current radiocarbon calibration curve, IntCal13 (Reimer et al., 2013), obtain their absolute age from climate wiggle-matching.' The climate wiggle-match records make up about 6% of the total data used in IntCal13 not 1/3 as stated (423 out of 7019 data points; IntCal13 database accessed 9 August 2018 http://intcal.qub.ac.uk/intcal13/ )

p. 7, lines 208-210 'The timescale of the Lake Suigetsu record has been inferred from matching its 14C record to the 14C variations in speleothems, additionally constrained by varve counting (Bronk Ramsey et al., 2012).' This statement seems a bit backwards to me since the varve counting provided the initial timescale which was then adjusted by matching the 14C records in speleothems, but if co-author CBR is happy with the way it's written then that is fine.

p.10, Figure 4. How are the 14C anomalies calculated here? Filtering is mentioned in line 292 but details are not given until section 3.4 and in section 4.3 where the error weighted mean is removed from the data for the Laschamp period. Obviously that was not the case for Figure 4. What do the dashed boxes represent?

Section 3.5 Change-point detection in climate records This is an abrupt shift from synchronizing 14C records and 10Be in ice core records to comparing to the timing or d18O shifts in climate records. The climate records considered are not even identified here except by a site name in Table 1. Presumably this should be part of Section 5 ?

Section 5. Figure 13. Why is the NGRIP Ca record used instead of d18O? A word of

explanation here would be useful.

p.24-25 line 722-723 'Since IntCal13 in principle should be tied to the U/Th-age scale. . ..'. This phrase needs some qualification since IntCal13 is tied to dendrochronological time scale for 0 to 14,000 cal BP and while the Hulu cave U-Th agrees well with the tree-ring data it only begins at 10,730 cal BP.

'Since IntCal13 in principle should be tied to the U/Th and dendrochronological age scale . . .. . .'

All figures would benefit from being presented in a larger size.

──────────────────────────────

---

## Referee Comment (RC4) · F. Parrenin (Referee) · 30 Aug 2018

This manuscript discusses the relative timing of DO events observed in Greenland ice cores with those observed in dated speleothems. The methodology is based on the synchronisation via cosmogenic radionuclides. The synchronisation is done during three intervals where variations in production of cosmogenic radionuclides are important: 11-13 ka, 21-23 ka and 41-43 ka (Laschamp event). In-between these three time periods, a kind of interpolation is done and its uncertainty is evaluated thanks to a statistical method which assumes the GICC05 MCE as age interval uncertainty. It

is found that DO events are synchronous in ice cores and speleothems within uncertainties (189 yr). Moreover, GICC05 is found to agree with the U-Th chronology of speleothems within its MCE uncertainty, although clearly the MCE is strongly correlated in some intervals (e.g. uncertain layers are always real layers).

This is an interesting manuscript which is very well written. I will focus on the discussion of chronologies since I am not an expert of cosmogenic radionuclides. The only main comment I have is that the title and the formulation of the manuscript are a bit misleading since this manuscript does NOT provide a continuous connection of ice core and speleothems chronologies, but rather a discrete one during only three time periods. The interpolation which is done in-between is just an interpolation and in my opinion should not be treated as a continuous synchronisation.

---

## Author Comment (AC1) · 3 Oct 2018

We kindly thank all reviewers for their insightful criticism that helped us to improve this manuscript. Below we reply to the review comments one by one. The review comments are shown in grey, our reply in black. Applied changes to the manuscript are shown in red.

**Reviewer #1: Niklas Boers**

Summary:

This paper provides a very thorough synchronization of the GICC05 time scale obtained from counting annual layers in ice cores, and (assumed to be) absolute U/Th dates from several (sub-)tropical speleothems via cosmogenic radionuclides with a focus on 14C, for the time period from 10ka to 45ka BP. Based on this synchronization, the timing of the DO events during this interval is compared among ice core and speleothem records, and it is concluded that on average, no systematic lead or lag can be inferred, given the inherent uncertainties.

The paper is written very well, the subject is of great scientific importance, and the employed methodology seems accurate to me. I hence strongly support publication of this study in CP.

Thank you!

However, there are some instances where the presentation is not detailed enough at least for me to be able to precisely follow what is done exactly (see specific comments below). In addition, I have some slight conceptual concerns that I would suggest to be addressed prior to publication. Please note that I'm not a geochemist, so I apologize in advance for potentially trivial or irrelevant questions / concerns below.

Major comments:

1. Necessarily, some of the uncertainties are put in 'by hand', such as treating the MCE of the GICC05 time scale as 1 sigma, but also at several instances of the analysis of the cosmogenic radionuclides. This is not a critique per se, and I agree with the authors that their uncertainty estimates are probably very conservative. However, in the situations at hand, it cannot be quantified _how_ conservative, and this leads to a tricky situation: the more conservative the error estimates are chosen along the way, the harder it is to reject the null hypothesis of synchronous DO events in the different records. The final sigma reported for the average over all DOs and speleothems is 189yr, and a lot can happen with such uncertainties; the statement in the conclusion that on average the DO onsets occurred synchronously is thus misleading, I think. I'd suggest to rather emphasize here that no systematic leads or lags can be inferred given the (partly subjectively introduced) error estimates.

We agree, that 189 years is unfortunately still a substantial uncertainty. Also with respect to the comment by reviewer #2 we have reformulated our manuscript in the respective sections to say: "we reject the hypothesis of leads or lags larger than 189 years at the one sigma level."

In addition, the 189yr are not far from the delay between NGRIP and WAIS inferred to be significant by the WAIS members, would you mind to comment on this?

We don't see a relation between our inferred uncertainty that mainly arises from uncertainty in matching 10Be and 14C records, and the delay of the Southern Ocean response to the bipolar seesaw. Note that our 189 years are an uncertainty (the best guess is 26 years), while fur Buizert et al. (2015) the best guess is 218±46 years. The delay inferred by Buizert et al. is possibly related to the time it takes for eddies to propagate the temperature anomalies related to the bipolar seesaw across the Antarctic circumpolar current (Pedro et al. 2018, QSR). We don't see a reason why this mechanism should be related to our uncertainty estimate.

2. It is stated in the abstract, introduction, and in the discussion that the GICC05 uncertainties are reduced by 50-70%, but I don't understand where these numbers come from, and as far as I can tell,

they are not mentioned / explained somewhere else in the manuscript. If you compare the GICC05 MCE to the sigma of the transfer function ensemble, it might be problematic, since the MCE is not really related to a normal distribution, despite the pragmatic approach by Andersen et al.

The 50-70% was indeed derived by comparing our 95.4% probability interval to the MCE (see for example figure 12) adopting the pragmatic approach of Andersen et al. to regard the MCE as a 2sigma uncertainty. But we agree that this quantitative comparison may not be ideal. Hence, we used the more qualitative formulation "strongly" instead.

3. It is not clear to me how exactly the interpolation in Sec. 4.4 is carried out. This is a key part of the study, and I would hence suggest to make this section considerably more detailed. It is written that the AR(1) realizations are used for interpolation, but how? You sample from the PDFs at the tie points, but how do you make sure that a given AR(1) realization, starting at one tie point, ends up close to the next tie point? Note that I might be completely off track here.

As we write in the manuscript, we use the AR-process only for interpolation. Thus, this is not a random walk that by itself ends up at the tie-point. It is forced to do so. We generate the AR-noise purely based on the MCE (see pp. 19, l. 551-553) and then anchor it at the sampled tie-points, by calculating the difference between the AR-process realization and the PDF samples at the tie-points, and linearly correcting the AR-noise for this offset. As a result, the AR-process realization will be forced to run through the sampled tie-point, but vary freely in between, which gives us our interpolation uncertainty.

Also with respect to the comments below, we see that apparently our description of the way how we infer our interpolation uncertainty is not clear. Hence, we rewrote the entire section (see below). However, as we also write in the paper want to also stress again, that this section is merely an attempt to infer a conservative interpolation uncertainty while still using some constraints GICC05 is giving us.

"To construct a continuous transfer function between GICC05 and the U/Th timescale we apply a Monte Carlo approach. Each iterations consists of i) randomly sampling the PDFs at each tie-point and ii) interpolating in between the tie-points using an AR-process that is constrained by the GICC05 maximum counting error (mce). We use the tie-points shown in figure 7, 9, and 10, i.e., three tie-points between ice cores and tree-rings during the deglaciation, one tie-point between ice cores and the combination of Corals, Speleothems and Lake Suigetsu during the LGM, and one tie-point between ice cores and the Bahamas speleothem around the Laschamp event. For the interpolation, we use the time derivative of the mce (i.e., its growth rate) as an incremental error estimate. During periods when the growth rate is > 0 GICC05 may be stretched (compressed), while a growth rate of 0 does not allow this, independent of what the absolute mce is at that time. By multiplying this growth rate with a random realization of an AR-process, we simulate how much of that uncertainty has been realized in each iteration of the Monte Carlo simulation. Subsequently integrating over the resulting timeseries of simulated miscounts, we obtain again an absolute error estimate, i.e., one possible realization of the mce. In each iteration, this realisation is then anchored at the sampled tie-points (step i) by linearly correcting the offset between the sampled tie-points and the simulated counting error. Hence, this procedure provides us with a correlated interpolation uncertainty over time, taking into account some of the constraints provided by the ice core timescale itself, but giving priority to our inferred tie-points. We note that the treatment of the mce as an AR-process leads to larger interpolation errors compared to assuming a white noise model, which would lead to very small uncertainties that average out over long time (see also discussion in Rasmussen et al., 2006). Furthermore, we treat the mce as ±1σ instead of ±2σ as proposed by Andersen et al. (2006) which additionally increases our interpolation error. We stress that this procedure does not aim to provide

a realistic model of the ice-core layer-counting process and its uncertainty which is clearly more complex (see Andersen et al., 2006; Rasmussen et al., 2006), nor should it be interpreted such that the mce was a 1σ uncertainty. However, our approach allows us to infer a conservative estimate of the interpolation uncertainty while at the same time it takes advantage of the fact that GICC05 is a layer counted timescale and hence, cannot be stretched/compressed outside realistic bounds. This procedure was repeated 300,000 times which was found sufficient to obtain a stationary solution, leading to 300,000 possible timescale transfer functions."

Specific comments:

p3, l83: How were the 50-70% uncertainty reduction inferred quantitatively?

As mentioned earlier, this is based on a direct comparison of the MCE and our 95.4% probability estimate. Regarding the earlier comment, we changed it to the less quantitative "strongly".

p3, l91: the "Hence" suggests that the previous sentence implies the _inverse_ relationship, but I don't think it does, although I don't question the inverse relationship itself.

Changed to "This causes the production rates of cosmogenic radionuclides to be inversely related…"

p6, l179: what do you mean by "more direct function of the timescale?"

As outlined on p6, L171-178, long term changes of accumulation rates depend on ice flow/thinning models of the ice sheet. On shorter timescales however, this thinning can be assumed to be near constant. In that case, accumulation rate variability depends only on the variability of the determined annual layer thickness, which is a direct product of the timescale that defines the age-depth relationship of the ice core.

We added on p6, L179:

"…more direct function of the timescale that determines the age-depth relationship and, thus, annual layer thickness, and is very precise…."

p6, l190ff: Using both flux-corrected and non-corrected version of the ice core records is fine to infer systematic differences between the records via comparison to the expected error of the mean, but I find it a bit problematic to use such a stack for the synchronization; do you obtain different results when using only flux-corrected or only non-corrected versions of the records?

We agree with this concern and in fact tested this during the analysis. All results shown in the manuscript are robust with respect to whether we chose just single ice cores or versions (flux/flux corrected) of the records. However, we do believe that stacking all ice cores increases the signal to noise ratio and thus, yields the best estimate.

p7 l215: please define "cal"

added: "(calibrated before present, AD 1950)"

p7 l221: Could you motivate the assumption of proportional 10Be and 14C production rate changes here?

We added "(see also following section)" on p7, L.221, where we deal with the question of 14C:10Be production rate ratios in great detail.

Fig.2: the time scales are not equidistant, right? How do you perform the the FFT filtering? Do you interpolate? If yes, using which method, and to which resolution?

Since Figure 2 shows modelled D14C data, the shown records are indeed equidistant, that is annual. The original ice core data are of lower resolution (between a few years to ~150 years). The ice cores

were sampled more or less continuously, so that each radionuclide sample integrates over a given depth/age interval. Hence, a 10Be sample is an average of the 10Be concentration (or production) over an interval. Consequently, when producing the ice core stack (section 3.1) we enter each core as a step function. Firstly, this is closest to how the data is sampled, and secondly, this is important because the carbon cycle integrates over production rate changes. Hence, it matters for how long a given production rate is sustained.

However, with respect to the original question, we note, that the sampling resolution of the raw ice core data is sufficiently high (median resolution between 25 years for GRIP 10Be, and 130 years for GISP2 10Be), that calculating a 5000 year high pass filter is not sensitive to the interpolation method.

p10, l315: It may be my fault, but where in the results section are the window length and frequencies given? Can you be more specific?

We believe that we give these details:

P14, L412: "(<1000 years)"

P14, L430-431: "All data are FFT-filtered to isolate D14C variations on timescales <1000 years"

P14, L433-434: " Each of the lower panels refers to a 2000-yer subsection of the data"

P15, L.444-445: "…we chose to linearly detrend each datset (instead of band-pass filtering)…"

P15, L.446-447: "we have to increase the length of the comparison window to 4,000 years"

P17, L.504-505: "For this tie-point, we merely remove the error-weighted mean between 39-45ka BP from each dataset, since detrending would remove the largest part of the signal".

However, we now provide the details also more clearly in the method section:

"For the highly resolved tree-ring data we use a 1000 year high-pass FFT filter, while the lower resolved and more unevenly sampled coral/speleothem/macrofossil data is filtered by linear detrending to avoid the interpolation to equidistant resolution required for FFT analysis. Similarly, the high sampling resolution of the tree-ring data allows us to compare the data in 2,000 year windows, while we increase the window length to 4,000 and 5,000 years for the lower resolved data prior to 14ka BP. The exact frequencies and window lengths are also given in the results section."

p10, l324: I don't understand this sentence: do you mean that the delay between associated peaks in different sinusoidal signals increases with wavelength? Why?

Correct, that is what we mean. This is a known effect arising from the convolution of the production signal by the carbon cycle, owing to the different reservoirs that have different exchange rates and isotopic signatures. See for example figure 5 in Roth and Joos 2013. As we write 3 lines above (p10, L321-323):

"D14C variations in the atmosphere are dampened and delayed compared to the causal production rate changes. Both factors, attenuation and delay, depend on the frequency of the production rate change (Roth and Joos 2013; Siegenthaler et al., 1980)."

p11, l328: is the box-diffusion model by Siegenthaler et al referred to here?

In section 3.3 ("Carbon Cycle modelling") we state that we're using the box-diffusion model by Siegenthaler et al. 1980 (P7, L219-220). However, the statement on P11, L328, is independent of which model is used – there will always be a delay between production rate driven changes of atmospheric and marine D14C due to the carbon cycle.

p13, l284f: what do you mean by "deviations from the transition"? I find it a bit problematic to refer the reader to a paper in preparation here, since it's not clear given the presentation here, how the change-point detection is carried out. In particular, further below it becomes clear that for each

potential change point, PDFs are obtained for its onset, mid point, and end point, but it's not clear how these PDFs are derived.

By "deviations from the transition" we mean the deviations from the fitted ramp, which we describe as AR(1) noise, compared to the stadial-interstadial transition which would be the "signal" in our case. The PDFs are generated using a MCMC sampler.

We rewrote the method section to hopefully be clearer:

We use a probabilistic model to detect the onset, mid-point, and end of the rapid climate transitions in each individual record. The employed model describes the abrupt changes as a linear transition between two constant states. Any variability due to the long-term fluctuations of the climate records around the transition model is described by an AR(1) process that is estimated in conjunction with the transition model. The model is independently fitted to windows of data on their individual timescales (Table 1 & Fig. 13) around the rapid transitions. Inference was performed using Markov Chain Monte Carl sampling (MCMC) to obtain PDFs of the timing of the onset, the length, and the amplitude of each transition in each record. Using these PDFs we can calculate delays of the onset, mid-point and end of the climate transitions between different records, propagating the respective uncertainties of the parameters. For each record, only events that are well expressed and measured in high resolution have been fitted. The approach and inference procedure are described in more detail in Erhardt et al. (submitted).

Fig.7: -there seems to remain quite a discrepancy between the variability of the bold grey line and the Towai treering data (green) even after synchronization, could you comment on this?

It is true, that the agreement is not perfect even after synchronization. Disagreements can arise from measurement noise, changes in 10Be transport and deposition, or carbon cycle changes. Without dedicated modelling, it is impossible to pinpoint the exact reason for individual discrepancies. These features also exist in the Holocene (see e.g., figure 10 in Adolphi et al. 2016, CP). However, 10Be and 14C agree well before (12-12.7 kaBP) and after (13.2-14.5kaBP) the disagreeing section around 13 kaBP. Given that the ice core timescale has small differential uncertainties, we find it unlikely that the disagreement in between 2 well matching sections can be due to errors in the timescale. Instead we think this highlights our use of relatively long windows to be compared, instead of peak-to-peak matching.

- if I'm not mistaken, none of the time scales of the shown data are equidistant, how to you do the FFT filtering in this case? If you interpolate, how?

As mentioned earlier, the annual resolution of the modelled D14C record arises from the step-function used for the carbon cycle model input. The resolution of the different ice core records is better than 50 years during this interval. The measured 14C data is decadal (tree-rings) to multi-decadal (speleothem). For the high-pass filter, we calculate an error-weighted mean of overlapping 14C data, interpolate annually, and calculate a 1000a low-pass filter. That low-pass filter is interpolated back to the original data resolution, and subtracted. Again, we note that all original data has a sampling resolution substantially higher than the cut-off frequency, so that the filtering is insensitive to the interpolation algorithm.

p15, l445: can you explain what you mean by "to remove offsets"? This also relates to l312 on p10; aren't offsets at longer time scales potentially problematic? I guess _heuristically_ these longer-term offsets are attributed to carbon cycle changes, but a clarifying sentence would be good, I think.

The different 14C records have partly systematic offsets between them, possibly due to reservoir (corals), dead-carbon (speleothems), or other (measurement) effects. Since we are only interested in relative changes, and not absolute values, we can remove those to isolate common co-variability.

We added (see figure 1i) to make clear that we are referring to differences between the different 14C records.

Regarding offsets between 10Be and 14C on longer timescales: We think we outline our reasoning clearly on p10, L309-311:

"For our analysis we employ high-frequency changes in D14C since carbon cycle changes have only limited effects on atmospheric D14C on shorter time scales (Adolphi and Muscheler (2016). Similarly, as shown in figure 2, the agreement of the different ice-core records is better on shorter timescales."

In addition we motivate this approach already in the introduction on p4, L137-142:

"It is currently not possible to quantitatively correct either of the radionuclides for these non-production influences since neither past carbon cycle changes nor atmospheric circulation changes are sufficiently well known. However, the potential amplitude of non-production rate changes can be assessed through sensitivity experiments and added as an uncertainty for the production rate signal (Adolphi and Muscheler, 2016; Köhler et al., 2006)."

Furthermore, we discuss on page 6, L 175-181, why we think also the long term trends of the ice core 10Be data have large systematic uncertainties. In this sense, removing the long term trend reduces these uncertainties as well.

Fig.8: You present the result of the synchronization, and show the PDFs for the different windows, but I think one or two extra sentences in Sec 3.4 on how exactly these PDFs are used to shift the record across the windows would be very beneficial.

We do not use the PDFs shown in figure 8 for the final synchronization. We rewrote section 4.4 (see earlier) which now reads more clearly:

"We use the tie-points shown in figure 7, 9, and 10, i.e., three tie-points between ice cores and tree-rings during the deglaciation, one tie-point between ice cores and the combination of Corals, Speleothems and Lake Suigetsu during the LGM, and one tie-point between ice cores and the Bahamas speleothem around the Laschamp event."

Furthermore, we added at the end of section 4.2 where the tie-point is presented L.520:
"We used this tie-point (figure 9) in the final synchronization as it is the best-defined feature in this time interval, and consistent within error with the estimates shown in figure 8."

Fig11: there's no inset and no blue dashed line!

We assume that this comment refers to p18, L537. This should of course read Fig. 10 instead and has been corrected.

Also, shouldn't the four individual speleothem dates correspond to the measured (black) points of NRM/ARM?

No, the U/Th dates have been carried out at different depths than the geomagnetic analyses.

p.19: As noted above, I don't understand how you interpolate between the tie points, the description is too brief in my opinion: by derivative of the MCE, do you mean the increments from one measured point to the next? Why do you multiply the AR(1) with these? Which "cumulative sum back in in time", i.e. from where to where? You say "strong autocorrelation", but what is the value of the parameter alpha?

We hope that we could clarify this in our earlier reply and the rewriting of that section.

p19, l 575: what do mean by "grow/shrink at a rate determined by the mce"? the latter is cumulative and hence always increasing back in time, but your AR(1) based uncertainties decrease when going back in time towards the next time point. I agree that it should decrease this way, but I don't understand the method sufficiently from the given description to understand how, specifically.

We hope we clarified this above. While the mce is typically plotted as a cumulative error back in time, it is really its growth rate that determines the counting error for each time interval.

p22, l639: Here you say that you sample the PDFs for the DO onsets; am I correct in assuming that for each onset, you obtain a PDF of its dates from the change-pointdetection?

Correct. We added "(section 3.5)"

p23, l679: I don't think that this study _shows_ that the counting error can be strongly correlated over extended period; please correct me if I'm wrong!

We do think that it shows exactly that. The results by Adolphi et al. (2016) show that the offset between the tree-ring timescale and GICC05 during the Holocene, requires that nearly every layer, that has been marked "uncertain", has in fact not been a year. Similarly, our results show that to reconcile GICC05 and tree-rings/speleothems between 10-22kaBP require that almost every uncertain year in this period has been a "real" year. Thus, we think that our statement is correct.

But it is true, that we do not derive this explicitly, so we changed it to: "implies".

p25, l727-738: it would be good, I think, to add reference on the relation to the ITCZ position already here.

To provide a theoretical reference to why the ITCZ may migrate in concert with North Hemisphere abrupt events we added a reference to Schneider et al. (2014) in line 731.

p25, l739ff: The fact that the precip increase in El Condor and Cueva del Diamante significantly predates the onset of H2 in Greenland suggests that the southward shift of the ITCZ, proposed to explain the precip increase, was not caused by H2, but rather by long-term solar insolation changes and in particular the NH minimum around this time, right? Also, Fig.16 suggests that the variability in AMOC strength (related to H2) does not substantially affect the position of the ITCZ, but merely the precipitation anomalies north and south of the ITCZ. If this is correct, please revise the paragraph accordingly.

We changed: "…during a weak AMOC state, reduced advection of moisture from the tropical Atlantic leads to lower precipitation north of the ITCZ, while the ITCZ position over South America itself changes very little (Fig. 16)."

And: "It is hence possible, that when northern hemisphere summer insolation reached its lowest values over the past 50 kaBP around H2, the ITCZ migrated to a position south of El Condor and Cueva del Diamante, and during its transition caused the reconstructed precipitation change."

p27, l783: see above regarding the 50-70%

See earlier reply. Changed to "strongly"

p27, l784: note the above comment on the formulation that DO events occur on average synchronously, rather, the null hypothesis of synchronicity cannot be rejected given the uncertainties. Your statement in the abstract is more accurate, I think.

See earlier reply. Changed to "We reject the hypothesis if leads or lags larger than 189 years between Greenland, East Asia, and South America at the one sigma level."

Sorry for the lengthy report, I hope it's helpful!

Best,

Niklas Boers

Thank you for providing this valuable input. We think it improved the manuscript!
* * *
Reviewer #2: Jeff Severinghaus

This paper addresses a crucial problem we face in paleoclimatology - namely that many of us are going ahead and using the U-Th-dated speleothems to improve other paleo chronologies, without really having answered the fundamental question of whether the abrupt DO events seen in speleothems are synchronous with those seen in Greenland ice cores. I am as guilty of this as anyone - in Buizert et al. (2015) Clim. Past, 11, 153–173 we made a physical argument based on known atmospheric and oceanic processes that the Chinese speleothem DO events cannot have lagged Greenland's DO events by more that 50 years. We then proceeded to tie the Greenland and WAIS Divide timescales in a pragmatic fashion to the Chinese speleothems, adopting an uncertainty of 50 years due to the assumption of synchroneity. I do believe that this argument is solid, but it is not enough for the high scientific standards we as a community must ultimately achieve, and the authors of the present paper are attempting to rectify this problem and empirically show that this lag cannot be very large. Therefore this work is essential, timely, and critical to the paleo field, and therefore I think this paper should be published with only very minor revisions.

Thank you!

The ultimate uncertainty that the authors arrive at is large, unfortunately, so it is perhaps best if the language of the conclusions is adjusted to reflect that large uncertainty. Instead of saying that the speleothems and ice cores are synchronous within uncertainty (which is true), it might be more helpful to the reader to write "we can reject the hypothesis of asynchrony larger than 189 yrs" or something equivalent. That way the conclusion shows what has actually been added by the present work.

Changed accordingly.

Minor comments:

The term "synchronicity" is used in psychology (i.e. Carl Jung) and has nothing to do with paleoclimate or chronology. The proper term is "synchroneity". Please change all the uses in the paper accordingly.

Done.
* * *
Reviewer#3: Paula Reimer

This manuscript uses cosmogenic isotopes to synchronize the Greenland ice core timescale with the U-Th timescale through a meticulous, multi-step process. The authors minimize the root mean square error in the production rate models from geomagnetic field based reconstructions and the ice cores to resolve the scaling factor for 10Be. They then compare 14C archives from around the Lachamps event with the reconstruction from the scaled ice core stack to select the most suitable ocean ventilation rate for the carbon cycle. The investigation into the effect of delay between ice core reconstructed atmospheric 14C changes and the marine and speleothem archives was insightful. Once the ice cores were synchronized to the U-Th (and dendrochronological) timescale the synchroneity of the proxy response to D-O cycles in a number of speleothem climate records was tested. This represents a very important step in interpretation of palaeoclimate records. The ice core based 14C reconstruction will also provide a guide to improvements for the next IntCal radiocarbon calibration curveupdate.

Thank you.

Specific comments: p. 2, line 52-54 'About one third of the data underlying the current radiocarbon calibration curve, IntCal13 (Reimer et al., 2013), obtain their absolute age from climate wiggle-matching.' The climate wiggle-match records make up about 6% of the total data used in IntCal13 not 1/3 as stated (423 out of 7019 data points; IntCal13 database accessed 9 August 2018 http://intcal.qub.ac.uk/intcal13/ )

We are sorry for this imprecision (in multiple ways).

Firstly, we are of course only referring to the glacial part older than 13.9kaBP where IntCal13 only consists of archives other than tree-rings, but which is also the period of the occurrence of DO-events, which is relevant for our discussion. In this section, 1623 14C determinations enter the curve of which 412 are climate wiggle-matched (Cariaco unvarved, Iberian Margin, Pakistan Margin). So that is 25%.

We clarified this in the manuscript:

"The current radiocarbon dating calibration curve (IntCal13, Reimer et al., 2013) is constructed from accurately dated tree-ring chronologies back to 13.9 ka BP (13.9 ka BP, kilo-years Before Present AD 1950). Beyond this time, which encompasses all DO-events, about one fourth of the data underlying IntCal13 obtain their absolute age from climate wiggle-matching."

p. 7, lines 208-210 'The timescale of the Lake Suigetsu record has been inferred from matching its 14C record to the 14C variations in speleothems, additionally constrained by varve counting (Bronk Ramsey et al., 2012).' This statement seems a bit backwards to me since the varve counting provided the initial timescale which was then adjusted by matching the 14C records in speleothems, but if co-author CBR is happy with the way it's written then that is fine.

We changed the statement to:

"The timescale of the Lake Suigetsu record is based on varve counting, corrected for long-term systematic errors by matching its 14C record to the 14C variations in speleothems (Bronk Ramsey et al., 2012)."

p.10, Figure 4. How are the 14C anomalies calculated here? Filtering is mentioned in line 292 but details are not given until section 3.4 and in section 4.3 where the error weighted mean is removed from the data for the Laschamp period. Obviously that was not the case for Figure 4. What do the dashed boxes represent?

We removed the error weighted mean prior to the Laschamp event from all datasets. We added to the caption of figure 4:

"All data are shown as anomalies to their error-weighted mean prior to the Laschamp event. i.e., the $\Delta^{14}$C increase. The dashed boxes encompass the time periods and $\Delta^{14}$C uncertainties (error of the error weighted mean) used for the definition of the pre-and post-Laschamp event levels."

Section 3.5: Change-point detection in climate records. This is an abrupt shift from synchronizing 14C records and 10Be in ice core records to comparing to the timing or d18O shifts in climate records. The climate records considered are not even identified here except by a site name in Table 1. Presumably this should be part of Section 5 ?

We agree that this is a relatively abrupt shift. However, we think that this should still be part of the method section 3. We added a short introductory paragraph to the section:

"To test the synchroneity of rapid climate changes, we compare the timing of DO-events seen in Greenland ice cores (Andersen et al., 2004), to a number of well-known U/Th dated speleothems that show DO-type variability from Hulu Cave (Cheng et al., 2016), Sofular Cave (Fleitmann et al., 2009), El Condor, and Cueva del Diamante (both Cheng et al., 2013b)."

Section 5. Figure 13. Why is the NGRIP Ca record used instead of d18O? A word of explanation here would be useful.

We added in section 5:

"We used the NGRIP Ca record (Bigler, 2004), that shows the largest signal to noise ratio across DO-events (compared to e.g., $\delta^{18}$O) making their identification more precise. In addition, the Ca aerosols originate from Asian dust sources (Svensson et al., 2000) and are thus, more directly related to Asian hydroclimate (Schüpbach et al., 2018) making them potentially more comparable to for example the Hulu cave record. Potential phasing differences between different climate proxies in the ice core are small compared to our synchronization uncertainties (Steffensen et al., 2008)."

p.24-25 line 722-723 'Since IntCal13 in principle should be tied to the U/Th-age scale'. This phrase needs some qualification since IntCal13 is tied to dendrochronological time scale for 0 to 14,000 cal BP and while the Hulu cave U-Th agrees well with the tree-ring data it only begins at 10,730 cal BP.

'Since IntCal13 in principle should be tied to the U/Th and dendrochronological age scale…'

True. To be more precise in our formulation we changed the sentence to:

"Since IntCal13 in principle should be tied to the U/Th-age scale for sections older than 13.9 ka BP, this implies either an…"

All figures would benefit from being presented in a larger size.

We hope CP takes care of this request during the layout/typesetting process.
* * *
Reviewer #4: Frederic Parrenin

This manuscript discusses the relative timing of DO events observed in Greenland ice cores with those observed in dated speleothems. The methodology is based on the synchronisation via cosmogenic radionuclides. The synchronisation is done during three intervals where variations in production of cosmogenic radionuclides are important: 11-13 ka, 21-23 ka and 41-43 ka (Laschamp event). In-between these three time periods, a kind of interpolation is done and its uncertainty is evaluated thanks to a statistical method which assumes the GICC05 MCE as age interval uncertainty. It is found that DO events are synchronous in ice cores and speleothems within uncertainties (189 yr). Moreover, GICC05 is found to agree with the U-Th chronology of speleothems within its MCE uncertainty, although clearly the MCE is strongly correlated in some intervals (e.g. uncertain layers are always real layers).

This is an interesting manuscript which is very well written. I will focus on the discussion of chronologies since I am not an expert of cosmogenic radionuclides. The only main comment I have is that the title and the formulation of the manuscript are a bit misleading since this manuscript does NOT provide a continuous connection of ice core and speleothems chronologies, but rather a

discrete one during only three time periods. The interpolation which is done in-between is just an interpolation and in my opinion should not be treated as a continuous synchronisation.

Thank you for your feedback. We are not sure how to comply with this request though. We want to remind the reviewer that eventually almost any synchronization method is based on more or less discrete tie-points between timescales (volcanoes, rapid CH4 changes, climate-wiggle matching). In between, there is always some sort of interpolation required, which obviously becomes more uncertain as the distance between the tie-points increases. How much more uncertain it becomes depends on whether we have prior information on the stratigraphy of the archives. We exploit this information from the layer counted ice core timescale telling us how this uncertainty is growing width depth/time between horizons.

We believe that we i) never state we would provide a continuous synchronization (but a continuous transfer-function), ii) clearly illustrate in text and figures, that this is only based on a few tie-points, iii) provide conservative interpolation errors by treating the mce as correlated and 1 instead of 2 sigma.

Obviously we hope that more tie-points can be established in the future as new data becomes available. But as we show in figure 12, out transfer function is consistent within error with the few independent tie-points that are available for testing our approach, during a period that is far away from our actual tie-points.

In summary, we hope that our results provide a test-bed for future studies and believe that given the current constraints, we provide the best-guess for the timescale difference between Greenland ice cores and U/Th dated speleothems without applying climate wiggle-matching and the underlying assumptions.

---

## Author Response (AR2)

Reply to Reviewer #2: Niklas Boer

Dear authors, thanks a lot for your detailed and comprehensive responses to my comments, which I suggest for publication, subject to a minor technical point:

For purposes of reproducibility, it would be good to provide the technical details of the AR process used in Sec. 4.4: I assume it is of order 1 (i.e., an AR(1) process)? Which values do you choose for the AR1-parameter and the sigma of the white noise part of the process? How do you motivate your choices, and how do different choices impact the final uncertainty estimates shown in Fig.12?

We added the parameters of the AR1 process on page 19, L. 573 (Phi = 0.9, sigma = 1). In addition, we elaborated on this choice on page 20, L. 576-580:

"The parameters for the AR-process were chosen so that the simulated realization of the mce explores the whole absolute counting error space, without frequently exceeding the permitted growth rate of the mce. A larger $\Phi$ would increase interpolation uncertainty, but also frequently violate the constraints of the layer count. A smaller $\Phi$ on the other hand, would decrease the uncertainty due to shorter decorrelation length (see also discussion in (Rasmussen et al., 2006))."

Very best,

Niklas Boers

We want to thank you again, for your comments which helped us to improve this manuscript.

[revised manuscript text omitted]